# LoRaWAN Network for Fire Monitoring in Rural Environments

**Sandra Sendra** [1,2,*], **Laura García** [2], **Jaime Lloret** [2], **Ignacio Bosch** [3] **and Roberto Vega-Rodríguez** [1]

[1] Department of Teoría de la Señal, Telemática y Comunicaciones (TSTC), Universidad de Granada, Periodista Daniel Saucedo Aranda s/n, 18014 Granada, Spain; robertovr@correo.ugr.es
[2] Instituto de Investigación para la Gestión Integrada de zonas Costeras (IGIC), Universitat Politècnica de València, Paraninf 1, 46730 Valencia, Spain; laugarg2@teleco.upv.es (L.G.); jlloret@dcom.upv.es (J.L.)
[3] Institute of Telecommunications and Multimedia Applications (iTEAM), Universitat Politècnica de València, Camino Vera s/n, 46022 Valencia, Spain; igbosroi@dcom.upv.es
[*] Correspondence: ssendra@ugr.es; Tel.: +34-696-322-321

**Abstract:** The number of forest fires that occurred in recent years in different parts of the world is causing increased concern in the population, as the consequences of these fires expand beyond the destruction of the ecosystem. However, with the proliferation of the Internet of Things (IoT) industry, solutions for early fire detection should be developed. The assessment of the fire risk of an area and the communication of this fact to the population could reduce the number of fires originated by accident or due to the carelessness of the users. This paper presents a low-cost network based on Long Range (LoRa) technology to autonomously evaluate the level of fire risk and the presence of a forest fire in rural areas. The system is comprised of several LoRa nodes with sensors to measure the temperature, relative humidity, wind speed and $CO_2$ of the environment. The data from the nodes is stored and processed in a The Things Network (TTN) server that sends the data to a website for the graphic visualization of the collected data. The system is tested in a real environment and, the results show that it is possible to cover a circular area of a radius of 4 km with a single gateway.

**Keywords:** Long Range (LoRa); LoRaWAN; fire detection; low-cost; Arduino; monitoring; sensors; wireless sensor networks (WSN), Dragino; The Things Network (TTN)

## 1. Introduction

Forest fires are continuous threats during summer seasons. In the decade from 2008 to 2018, there was an average of ≈70,000 fires per year in the regions of Europe, the Middle East, and North Africa [1]. In 2018, the number of forest fires in Portugal, Spain, France, Italy, and Greece, which are some of the countries most affected by wildfires, was 26,434 [1]. These fires cause insecurity and large amount of material and, sometimes, even personal damage. Furthermore, the environment is affected by the $CO_2$ emitted by the fires and the amount of time required for reforestation may expand for more than 60 years [2]. Lightning strikes, accidents, negligence acts, and intentional acts are the principal causes of forest fires in several of these countries [3]. Figure 1 shows the number of wildfires in Europe, the Middle East, and North Africa in 2018 [1], while Figure 2 shows the burned hectares by wildfires in Europe, the Middle East, and North Africa in 2018 [1].

Both prevention and extinction measures are part of the plan against forest fires designed by the government of Spain. The preventive measures include specialized teams that evaluate the state of the forests and propose different actions to improve it, such as upkeeping the firebreaks.

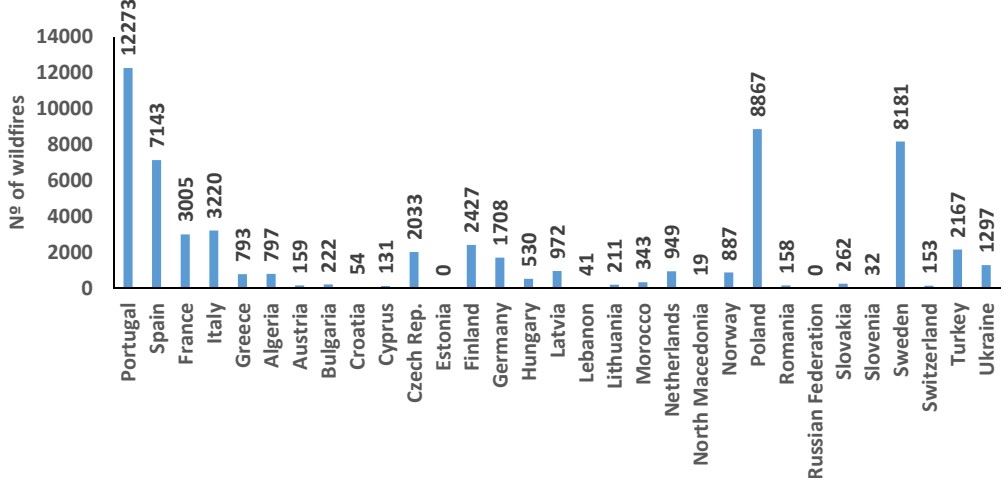

**Figure 1.** Number of wildfires in Europe, the Middle East, and North Africa in 2018.

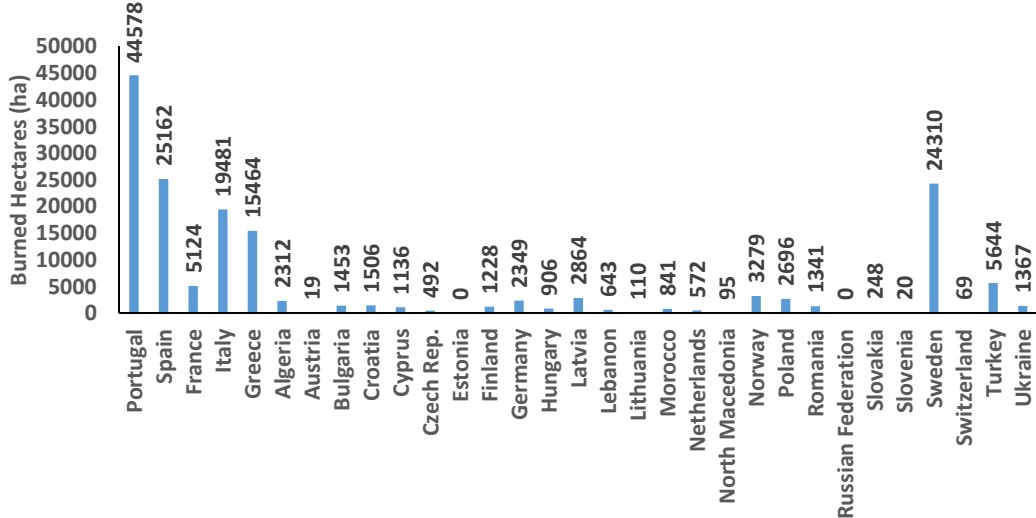

**Figure 2.** Burned hectares by wildfires in Europe, the Middle East, and North Africa in 2018.

The fire extinction measures include the use of aerial vehicles, the use of the SIMIF (Meteorological Information System for Forest Fires, originally in Spanish, *"Sistema de Información Meteorológica para Incendios Forestales"*) system for environmental monitoring regarding forest fires, which includes prediction and fire alert functionalities, GPS, data and, voice communication systems for the deployed forces and the use of satellite images. Some regions developed their fire detection and prevention system such as the SIGIF (Integrated Forest Fire Management System, originally in Spanish, *"Sistema Integrado de Gestión de Incendios Forestales"*) provided by the Generalitat Valenciana (Spanish Government) [4]. It provides information on the environment from the Spanish Meteorology Agency (AEMET) and the meteorological observatories of the region. It also shows the areas with emergency alerts and the locations where lightning landed.

Several environmental parameters are evaluated so as to identify the risks of new fires and prevent them. Temperature, humidity, precipitations and, wind speed are some of the commonly monitored parameters. These parameters are considered to produce an index that indicates the risk of forest fires, such as the Haines index [5]. This index determines with an integer from 2 to 6 the likelihood of a fire behaving in an erratic manner or becoming large. Moisture and stability are evaluated and assigned a value between 1 and 3. The sum of those values results in the Haines index, where value 2 is related to low potential of fire growth or erratic behavior and, the value 6 indicates a high potential.

Deploying WSN on forest terrains may present some difficulties. The absence of power sources leads to the need for using batteries and alternative power sources such as solar panels. Lower-range wireless technologies increase the cost of the system due to the need for a bigger infrastructure. Nonetheless, technologies such as WiFi, Bluetooth or cellular communications can be an option for fire detection systems meant to be deployed closer to inhabited areas such as rural environments [6]. Due to the scale of the areas that need to be covered, long-range wireless technologies are the best solution. LoRa presents itself as a good solution for this type of application as its range is up to several kilometers and it presents a low power consumption. Other LPWAN (Low Power Wide Area Network) technologies such as SigFox, LTE or, 5G could also be applied to WSN systems for rural and forest areas [7]. However, LoRa is gaining popularity and the availability of low-cost LoRa nodes and gateways is increasing, leading to the increment in its attraction for the development of applications that require the wireless coverage of large areas with difficult access. Additionally, compared to SigFox, which is the closest technology, LoRa can reach a higher Data Transfer Rate.

Considering the importance of early forest fire detection and the advantages of using LoRa in WSN-based forest fire detection systems, this paper presents a network for estimating fire risk on an area by using the 30-30-30 rule. This rule permits determining the level of fire risk in an area considering the values of relative humidity, wind speed, and temperature. A high fire risk implies having a temperature greater than 30 °C, a relative humidity lower than 30%, wind speed greater than 30 km/h and absence of precipitation in the last 30 days.

Additionally, it can also work as a fire detection system for the cases where intentional (or due to natural causes) fires were generated. The system is comprised of $CO_2$, temperature, humidity and, wind speed sensors that monitor the environment connected to an Arduino and Dragino LoRa Shield module. LoRa is used to send the information from different nodes located at the mountain to the gateway, where the data is sent to the TTN server to be processed by the application server and to graphically be shown on a website. The system was tested in a real environment. The tests are focused on the network performance, measuring the Received Signal Strength Indicator (RSSI), Signal-to-Noise Ratio (SNR) and, the operation of the sensors.

The rest of the paper is organized as follows. Section 2 presents the Related Work on other existing forest fire detection systems. An overview of LoRa and LoRaWAN is provided in Section 3. Section 4 presents the system description. The configuration of the TTN server and the design of the application server are detailed in Section 5. Tests and results are shown in Section 6. Lastly, the conclusion and future work are presented in Section 7.

## 2. Related Work

This section presents the current available WSN-based systems for fire detection.

Ahmad Alkhatib presented in [8] a forest fire detection WSN system that used a sub-network coverage method. The deployed network was comprised of Waspmotes that transmitted using XBee 802.15.4. The proposed method divided the network into three subnetworks which resulted in an increase of 2.7% of the lifetime of the network. Compared to other WSN deployments for fire detection, the proposed deployment obtained an increase of 63% of the energy performance.

Jorge Granda et al. presented in [9] a WSN system for forest fire detection intended to be deployed on the Guanguilatgua Metropolitan Park of Quito. $CO_2$, CO, and $CH_4$ gasses were monitored in real time with the MG-811, MQ7, and MQ4 sensors, respectively. Furthermore, temperature and humidity were monitored using the DHT22 sensor. The node was an ATmega328P Arduino that incorporated a NEO-6M GPS to provide the exact location of the measures and an XBee S2B Pro module to establish the communication between the nodes using the ZigBee protocol. The data could be visualized using a user interface created in C# in Visual Studio 2012 and, the data was stored in a SQL database.

Anton Herutomo et al. performed in [10] a reliability test on a Machine-to-Machine (M2M) forest fire WSN detection system. The authors employed the OpenMTC platform to carry out the Internet of Things (IoT) and M2M communication. The Arduino UNO nodes were comprised of an MQ7 CO gas

sensor, an LM35 temperature sensor and, a DHT11 humidity sensor, a GPS shield and an XBee adapter to enable the communication with ZigBee. The maximum range of the nodes resulted in be 341 m obtaining a transmission time of 1250 ms.

Miguel Antunes et al. presented in [11] a low-cost early wildfire detection system that employed LoRaWAN as the communication protocol. The system was intended for the protection of localized areas such as camping sites, villages or, isolated homes susceptible to being affected by the fire. Strong heat sources were identified using narrow-beam FIR (Far Infrared) sensors with a detection range of up to 50 m. These sensors were connected to STM32 L432KC nodes functioning in a master-slave manner. The master node received the temperature data and forwarded said data and the alerts in real time through an Internet connection. The proposed system was used along with countermeasures such as sprinklers so as to defend the areas of interest. These sprinklers were activated with latching valves that are able to maintain their status even with a power loss.

Antonio Molina-Pico et al. presented in [12] a hierarchical WSN for forest monitoring and a fire detection system. The nodes had a PIC24FJ256GB110 processor unit, an AC4868-250 module for communications on the 868-870 MHz ISM band and a GSM/GPRS module for cellular communication in case it was available. The GPS position of each node was collected when the nodes were deployed. The sensor nodes monitored CO, $CO_2$, temperature, humidity, wind speed, and wind direction. Tests were performed employing two simulators considering communication systems, fire brigades and coordination of aerial and land means.

Moumita Ghosh et al. proposed in [13] a three-dimensional multi-sink WSN scheme for forest fire detection. The scheme included an event-driven mode, a time-mode, and a hybrid-mode. The authors proposed an energy-efficient protocol that prioritized fire detection to energy-saving and used a Fermat point scheme. The results showed that the hybrid-mode presented the highest lifetime, followed by the event-driven mode and the time-driven mode whose lifetime was depleted faster due to using a random transmission mode instead of a round-robin mode and the transmission of unnecessary data respectively.

Saeed Ullah Jan et al. presented in [14] a WSN-based forest fire alerting system. The system was aimed at determining the scale and intensity of the fire. The authors proposed a dynamic routing solution and a data communication solution that incorporated energy-efficient and priority-based techniques. The Fire Weather Index (FWI), the energy, the security, and the weight were the parameters evaluated to form the dynamic paths. The client application provided information on environmental parameters such as temperature, humidity, rain, and wind speed. Said parameters were used to calculate the FWI. The areas with high FWI values were colored to mark the location of the fire. When fires were detected, an alert message was forwarded to the officers. Simulations were performed using a Microsoft Framework. The results showed the efficiency and the dependability of their proposal obtaining the same efficacy in detecting fires while reducing the consumed energy and the overhead of the data packets.

Emmanuel Lule et al. presented in [15] a WSN-based architecture for fire disaster monitoring in developing countries. Simulations were performed using the OPNET Modeler Ver. 14.0. AODV and DSR were compared for the case of mobile nodes moving at a speed of 10 m/s. The results showed that AODV obtained the minimum delay, 0.2 ms, and the maximum throughput, 1.7 Mbps. The authors concluded that AODV was better for larger areas whereas DSR was better for smaller areas. The authors then proposed a hybrid AODV+DSR mechanism to improve the performance compared to DSR. However, AODV+DSR did not outperform AODV.

Teng Ma et al. presented in [16] a mixed WSN for the forest fire monitoring paradigm (FFMP). The mobile nodes served as cluster heads that gathered the information from the static nodes. The cluster nodes fused the gathered data to send it to the base station. The data was then processed to create temperature graphs and to detect the fire. Simulations were performed to ensure the good performance of the proposal. Results showed an increase in the lifetime of the WSN.

Iván Froiz-Míguez et al. presented a monitoring system for industrial workers to evaluate their safety and health in real time [17]. Sensors were deployed on IoT wearables that communicated with the closest gateway using LoRaWAN and LPWAN technology. The collected information was stored in distributed locations and processed using blockchain to provide immutability and traceability to the data when sharing it with medical facilities or insurance companies.

Boris Benites et al. proposed [18] a water scarcity monitoring system intended for arid and semi-arid places. Humidity sensors and anemometers were incorporated to swarm drones to monitor zones with high concentrations of water. These drones forwarded the data using LoRaWAN to the registry for its storage. Then, the data would be transmitted to the central node to perform predict wind direction using classification and regression techniques.

Emiliano Sisinni et al. designed [19] a range extender for LoRaWAN industrial IoT. The authors suggested the use of a frame relay approach to avert compromising the highest data rates to higher sensitivity. The prototype was created from commercial hardware. Coverage tests were performed measuring the RSSI and the SNR to ensure the correct performance of the proposal and to demonstrate its effectiveness in increasing the range. Furthermore, the range extender was able to perform with legacy networks of LoRaWAN.

Previous works established wireless sensor networks using different communication protocols. LoRa is a new technology that has not been thoroughly tested for this purpose and the available studies on the usage of this technology for fire detection systems do not provide much information on the performance of LoRa in this kind of setting. This paper presents a WSN-based forest fire detection system that uses LoRa to establish long-distance wireless communication. Additionally, the system uses the 30-30-30 rule to determine if there is a high level of fire risk. So our system is able to both predict and detect fires. Finally, real tests were performed to determine the performance of the systems and the characteristics of the communication link on a real scenario.

## 3. LoRa Technology Overview

Long Range (LoRa) is a wireless technology in which a low power transmitter forwards small packets of data (between 0.3 kbps and 5.5 kbps) to a receiver, usually over a long distance. These features make LoRa fall into the category of LPWAN networks [20]. Although LoRa's coverage range is wide, it is highly dependent on the environment and the building materials that make it up. The range for rural surroundings is approximately 20 km. However, in urban surroundings, it is reduced to 5 km. Nonetheless, some tests confirm that data was received from a distance of hundreds of kilometers with adequate and unobstructed direct vision in the Fresnel area between the devices [18,21].

LoRa Wireless technology was developed by the French company Cycleo. In 2012 the American company Semtech acquired Cycleo, who now owns the patent for the radio part and the LoRa modulation and whose codes are closed. Semtech provides intellectual property licenses to other companies, especially to component manufacturers such as HopeRF, Microchip, etc.

On the other hand, LoRaWAN protocols are open and defined by the LoRa Alliance, a non-profit organization founded in 2015 with more than 500 partner companies (IBM, Microchip, Orange, Cisco, etc.) created with the commitment to allow and encourage large-scale deployments of LPWAN IoT devices that implement their standards.

This section presents an overview of this technology and the main characteristics that make it the most suitable to develop our proposal.

### 3.1. LoRa Modulation

Modulation is the way in which information (digital or analog) is encoded in the carrier signal that will be responsible for transmitting it. LoRa modulation is based on CSS (Chirp Spread Spectrum) which is a spread spectrum technique that uses linear frequency modulation chirp pulses with high bandwidth to encode the information. A chirp pulse is nothing more than a sinusoidal signal in which the frequency increases (up-chirp) or decreases (down-chirp) over time. A chirp determines a symbol.

The chirps change frequency for a certain amount of time, called symbol time ($T_s$). These "jumps" of frequency determine how the information is encoded. The number of bits that can be encoded in each symbol is given by the spreading factor (*SF*), so a symbol can have $2^{SF}$ values to jump to. These values are called chirps (e.g., if *SF* it is 7, the number of bits that the symbol can encode is also 7, and it has $2^7$ chirps). LoRa admits a range of integer values of *SF* between 7 and 12 [22].

The symbol rate $R_s$ is defined, as indicated in [23], as:

$$R_s(symbol/s) = \frac{BW}{2^{SF}} \tag{1}$$

Since the chirp rate is constant for a given bandwidth ($R_c = BW$), (1) can be rewritten as (2):

$$R_s = \frac{R_c}{2^{SF}} \tag{2}$$

Bit rate ($R_b$) is defined as:

$$R_b(bps) = \frac{SF{\cdot}BW}{2^{SF}} \frac{4}{4+CR} \tag{3}$$

where *CR* (Coding Rate) is the proportion of non-redundant bits for Forward Error Correction (*FEC*). LoRa modulation allows different values for this:

$$CR = \frac{4}{4+n} \tag{4}$$

where *n* = 1, 2, 3 and 4, so the allowed values are 4/5, 4/6, 4/7 and 4/8.

The duration (in seconds) can be calculated as $T_c(s) = \frac{1}{BW}$, since, as we previously stated, $R_c = BW$. The duration of the symbols is then defined by:

$$T_s(s) = \frac{2^{SF}}{BW} \tag{5}$$

As equations show, the bigger the SF value, the lower the bit rate and, the greater the symbol time. Therefore, it will increase the message transmission time or Time on Air (*ToA*), and the power consumption in the device transmitter. On the other hand, the coverage range will be larger at a higher SF value due to the resulting greater robustness against noise [24].

Regarding the use of different channels, signals received with different SFs are generally considered purely orthogonal. However, it is not true under certain power level conditions. The end node parameters (SF and transmit power) can be adjusted based on the distance from the gateway and, it offers the possibility to run networks with multiple gateways [25–27].

Finally, another characteristic of the LoRa modulation is its immunity against the Doppler Effect. The displacement caused by the Doppler Effect causes a small frequency shift to the modulated signal that hardly affects the baseband signal in the time domain.

### 3.2. LoRaWAN

LoRaWAN is a network protocol designed for LPWAN applications (i.e., requiring a wide range of coverage and low power consumption, in which usually the end devices are powered by batteries) that specifies the medium access control layer (MAC) and the application layer. The latest version of the protocol released by the LoRa Alliance is version 1.1 [28] that guarantees backward compatibility with its predecessor 1.0, the first version of the protocol.

#### 3.2.1. Network Architecture

The typical architecture of a LoRaWAN [29] network is based on a star topology (see Figure 3), in which there is a gateway (gateway) that forwards messages between an end device and a central network server. The network server directs the packets of each device in the network to an application

server. To provide integrity and confidentiality to the network, the protocol relies on a symmetric model of session keys derived from keys associated with each device (root keys). The keys associated with the devices and the session keys derived from them are stored by the incoming server (Join Server). The network server, the application server, and the Join Server can be hosted on the same machine.

The gateways are connected to the network server through the conventional TCP/IP SSL network while the end devices, using LoRa or FSK modulation, use LoRaWAN and communicate with one or more gateways, the latter being able to handle up to 6000 devices. The gateway has the function of collecting data from the different motes included in the network and connecting them to the rest of the network. In this process, three different flows can be observed [30]:

- Downlink flow: the router receives downlink messages from upstream and waits for a gateway near the device to be available. Finally, it builds downlink options from the gateway options and transmits it.
- Uplink flow: the router receives uplink messages from gateways and parses the uplink content for MAC payload and activations. Finally, it computes new downlink windows, finds a broker to transmit the packet to, and transmits the uplink.
- Status messages: the router uses these messages to track the active gateways.

To generate all these messages and transmit from a gateway to TTN, two different protocols, i.e., the legacy Semtech UDP protocol and the Gateway Connector protocol, are required.

The Semtech UDP protocol, uplinks, statuses, and downlinks are exchanged in a pseudo-JSON format between the gateway and the network server. The UDP protocol does not provide authentication and, there is no encryption available. For this reason, gateway messages can be intercepted during transport.

Using the gateway connector protocol, it is possible to provide security to the system. Gateways are identified by an ID and a key. Sending a message to a router requires knowing the correct ID/key combination. With the gateway connector protocol, messages can be exchanged through network protocols such as MQTT/MQTTS or using gRPC (that supports TLS encryption natively), if the hardware and software support it.

LoRaWAN does not support communication between the end nodes. However, it is possible to combine LoRaWAN with other specifications that do support it. In general, all communications between the devices are bidirectional.

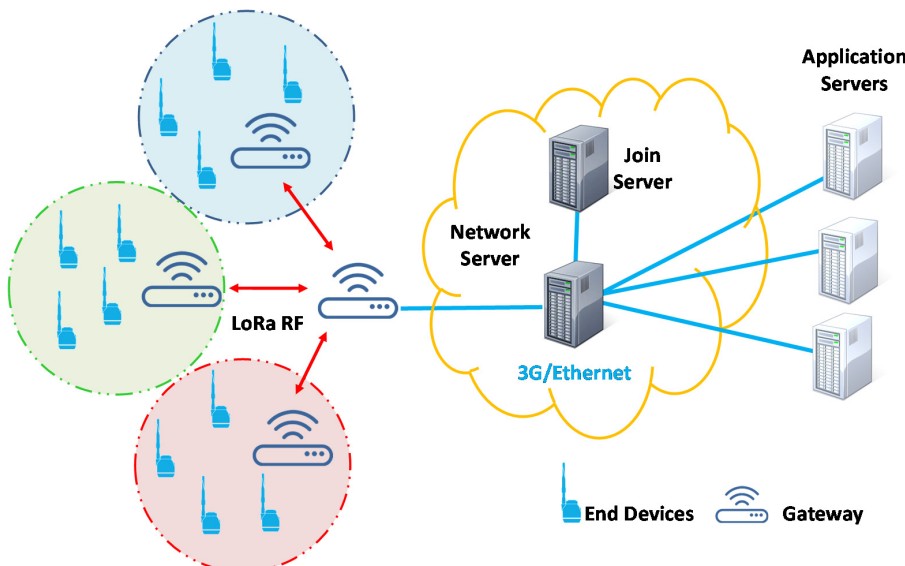

**Figure 3.** LoRaWAN network architecture.

An end device consists of two parts:

- A radio module with an antenna.
- A microprocessor for data processing (e.g., data from a sensor).

As mentioned before, given their small consumption, the end devices are powered by batteries. If an end device has sensors and acts as a remote sensor, it is usually called a mote.

A gateway also has two parts:

- A radio module with an antenna.
- A microprocessor.

The gateways are powered by the power grid and connected to the TCP/IP network. In addition, they can usually listen on different frequencies and each of the SFs of each frequency simultaneously.

### 3.2.2. Regulation

LoRa operates in the band without an ISM license that is available worldwide. In Europe, the frequency bands licensed for this use are the EU433 (433.05–434.79 MHz) and the EU863-870 (863–870 MHz) bands. In the USA, it is the US902-928 (902–928 MHz) band. This band has the main advantage of being free to use without a license but, due to its great use, there is a lot of interference and, the transmission rate is low. These bands are regulated by different organizations. In Europe, it is regulated by the European Telecommunications Institute (ETSI). In the USA, by the Federal Communications Commission (FCC). Most countries copy the rules standardized by these agencies except for some countries such as Japan and South Korea that have their own agencies.

In Spain, the Secretary of State for Telecommunications and the Information Society is responsible for applying the European standards set by the ETSI. Some of these rules are:

- Using the EU863-870 frequency band.
- For the uplink, the maximum transmission power allowed is 25 mW (14 dBm).
- For the downlink (at 869.525 MHz), the maximum transmission power is limited to 0.5 W (27 dBm).
- The duty cycle is 0.1% and 1%, depending on the channel.
- The maximum antenna gain is set to +2.5 dBi.

The duty cycle is a measure to reduce collisions in the band. LoRa bases its standard on a duty cycle of 1% in Europe [31]. This means that both end devices and gateways can only transmit 1% of the time (e.g., if there is a 500 ms ToA, a message can be sent again in 99 × 500 = 49,500 ms, 49.5 s). Figure 4 shows the value of Time on the Air (ToA) in ms for a bandwidth of 125 kHz, at the 863–870 MHz frequency band, using different payloads. It is worth mentioning that service operators may also have their restrictions of use.

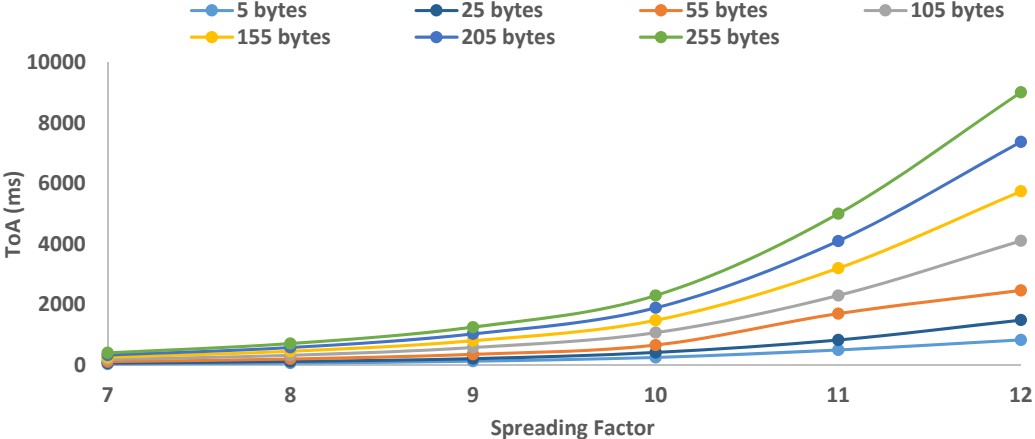

**Figure 4.** ToA (in ms) for different SF using a bandwidth of 125 kHz.

To compare LoRa with the rest of the IoT technologies, a comparison with the protocols Wifi [32], ZigBee [32], SigFox [33] and NB-IoT [33] is performed and showed in Table 1.

Some IoT applications [34], such as the one presented in this paper, have specific requirements such as long-range, low bit rate, and low power consumption. Solutions based on short-range technologies (e.g., Zigbee) or personal areas are not adequate for these types of long-range environments, which is why the Zigbee and Wifi standards are discarded for our purpose. If we focus on the LPWAN standards that do meet these requirements, we see that NB-IoT depends either on already deployed 3GPP networks or on its high-cost network deployment. It also has a more limited scope than SigFox and LoRa. As for technical specifications, SigFox and LoRa are similar. However, LoRa has the advantage of having an open protocol and, it allows the deployment of its own low-cost network unlike SigFox, in which we depend on the coverage offered by the operator. This is why LoRa was chosen as the protocol to develop the project. Figure 5 shows a comparison of the main features of the most used wireless technologies [32,33,35–37].

**Table 1.** Comparison of technologies used in IoT.

| | LoRa | Wifi | ZigBee | SigFox | NB-IoT |
|---|---|---|---|---|---|
| Frequency | 868 MHz (EU); 915 MHz (USA); 433 MHz (Asia) | 2.4 GHz and 5 GHz | 868 MHz (EU); 915 MHz (USA); 433 MHz (Asia); 2.4 GHz | 868 MHz (EU); 915 MHz (USA), 433 MHz (Asia) | Depends on the frequency licensed to LTE |
| Standard | IEEE802.15.4g, LoRa Alliance | IEEE802.11 | IEEE802.15.4 | SigFox (Owner) | 3GPP Standard |
| Coverage | 5 km (urban),20 km (rural) | 50 m (indoor), 40 km (outdoor, as a function of the visibility) | 10–100 m | 10 km (urban), 40 km (rural) | 1 km (urban), 10 km (rural) |
| Modulation | LoRa, FSK, GFSK | BPSK, QPSK, 16 QAM, 64 QAM, 256 QAM, 1024 QAM | BPSK, OQPSK | BPSK, GFSK | QPSK, OFDM (DL, SC-FDMA (UL) |
| Power consumption | Low | High | Medium-Low | Low | Low |
| Theoretical Data Transfer Rate | 22 kbps (LoRa), 100 kbps (GFSK) | 2.4 Gbps (IEEE802.11 ax, 2 streams with 1024 QAM) | 250 kbps at 2.4 GHz, 20 kbps at 868 MHz, 40 kbps at 915 MHz | 100 bps | 10 Mbps |
| Price of end devices | 3–5 € | 3–5 € | 2–5 € | >2 € | >20 € |
| **Price of Gateway/ Base Station** | 100 € Gateway/ >1000 € Base station | 20–600 € Gateway | 40–1000 € Gateway | 4000 € Base station | 15000 € Base station |

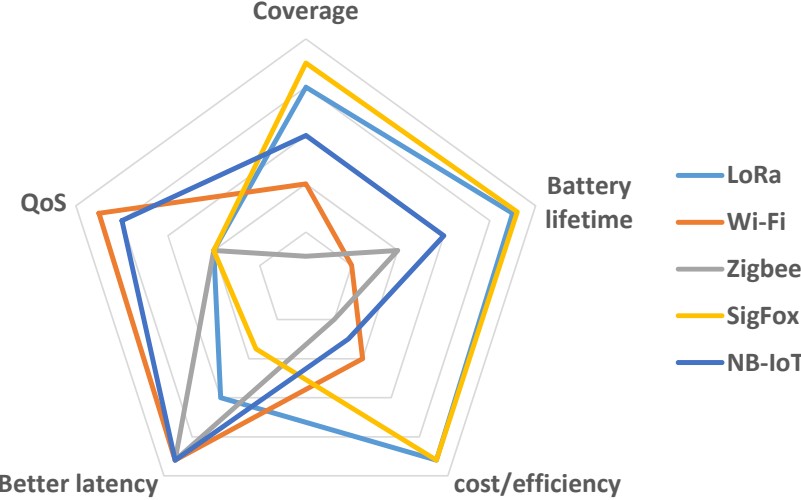

**Figure 5.** Main features of these technologies.

## 4. System Description

This section details the different elements our system is composed. Firstly, the network architecture developed to collect data from the environment is presented. Our system is comprised of an Arduino node with a LoRa Shield and several sensors. Finally, a LoRa Gateway is in charge of collecting data and sending it to our server to display it in a web interface.

### 4.1. Architecture

The network is based on LoRa technology, i.e., the network topology used in this case will be an infrastructure architecture where a gateway will be in charge of collecting data from nodes and, several nodes will be spread along within the scenario.

To implement our system, it is necessary to design and develop the end devices using the LoRaclass A specification [7] since it is the option that consumes less energy. The LoRa Gateway will be used to collect data from the LoRa nodes and to forward them to a server for storing and/or processing the data from the sensors.

The data will be stored in a network storage server. In this case, an already developed service provided by The Things Network (TTN) was used. It offers the necessary tools to collect and store the data in a database (DB) with the minimum cost.

Finally, a web application will be the user's interface to visualize the data from sensors and for the estimations of warning areas. Figure 6 shows the network architecture of our proposal.

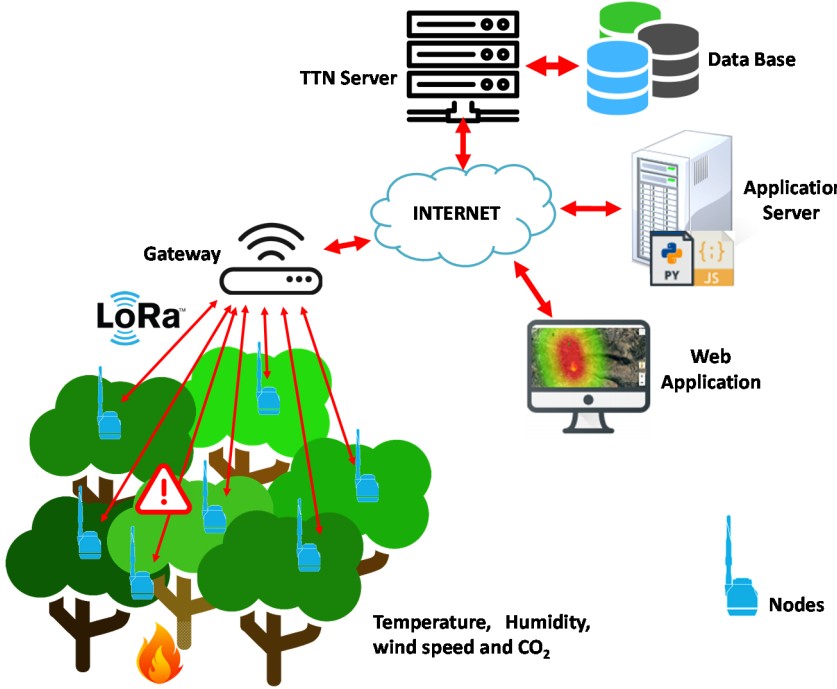

**Figure 6.** Network architecture of our proposal.

### 4.2. LoRa Node

The LoRa end device is in charge of gathering data from the environment through the different used sensors. The fire risk index estimation for an area that uses our system is based on the 30-30-30 rule. To do that, the node must gather information on temperature, relative humidity, wind speed and, rain of the last 30 days. The compliance of this rule and consequently, the existence of fire risk in an area, implies having a temperature higher than 30 °C, a relative humidity lower than 30%, wind speed higher than 30 km/h and absence of precipitation in the last 30 days.

According to [38], processing only two variables and checking the compliance of the limit values is enough to establish the warning of a fire in an area with a success rate of 70%. According to this study, for the 2007–2016 period, the compliance of the three variables specifically correlated with 40% of large fires. Since the rain is not used in this study, the $CO_2$ concentration was added as a method for verifying the presence of a fire. Finally, it is important to keep in mind that the occurrence of forest fires, sometimes, depends on the area where it is located and, its topography and therefore, parameters such as rain may not be important.

Figure 7 shows the block diagram of our nodes. As can be seen, the system is comprised of three sensors that measure four different parameters. Additionally, it will be powered by a solar panel and a battery.

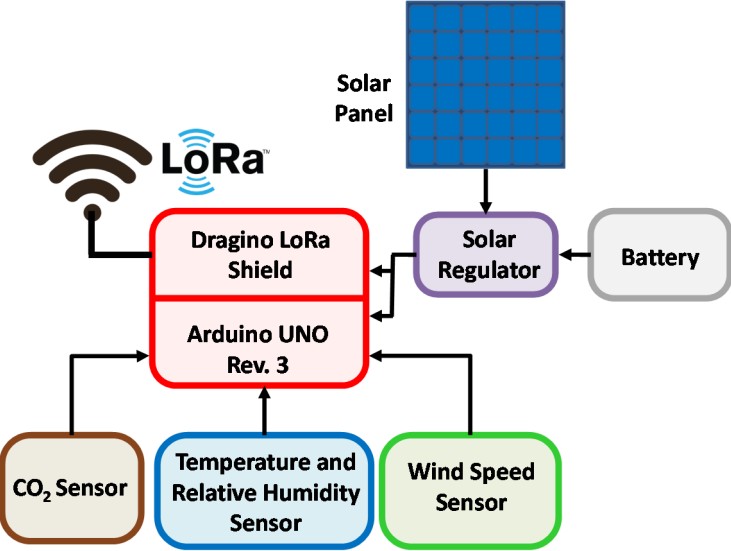

**Figure 7.** Block Diagram of node components.

The Arduino UNO Rev3 development board incorporates an ATmega328P microchip microcontroller [39]. To connect the Arduino node to the LoRa network, a Dragino LoRa Shield board [40] compatible with our Arduino development board is used. The Dragino LoRa Shield board incorporates the RFM95W transceiver [40] with the SX1276 chip (manufactured by Semtech) which allows using the LoRa modulation with the three working frequencies specified on the regulation, i.e., 915 MHz, 868 MHz, and 433 MHz [31]. Our model is pre-configured to work in the 868 MHz frequency band. Figure 8 shows the Dragino LoRa Shield coupled with the Arduino UNO module.

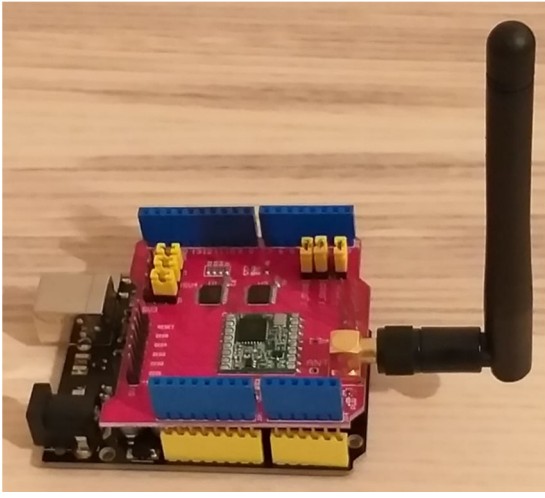

**Figure 8.** LoRa Node components.

### 4.3. Sensors

To collect data from the environment, it is required to add as many sensors as parameters are required to measure. This LoRa node is able to measure temperature, relative humidity, wind speed, and $CO_2$ concentration.

Firstly, to measure temperature and relative humidity, the system includes a DHT11 sensor [41]. This sensor offers a digital output due to a small 8-bit microcontroller which is already calibrated by the factory. The DHT11 sensor is able to measure temperature in the range of 0 °C to 50 °C with an error of ± 2 °C and relative humidity in the range of 20% to 90% with an error of ± 5%.

To collect the data of the $CO_2$ concentration in air, an MQ135 sensor is added [41]. It also includes a small microcontroller that generates the output values in both digital and analog formats. The sensor can detect NOx, NH3, different alcohols, benzene, smoke, and $CO_2$. The manufacturer recommends the load resistance to range between 10 kΩ and 47 kΩ. Its operation depends on a chemical reaction that takes place inside the sensor through a resistance that heats up. The measured parameter is related to the resistive value of this heated resistance which is measured at the output of the circuit after driving through the load resistance. Finally, this sensor requires a pre-heating phase of 24 h.

Therefore, following the recommendations of the manufacturer to correctly calibrate the sensor, the sensor has to keep running for 24 h. After that, the output resistance should be determined by using the library "RZero" of Arduino. $R_{Zero}$ or $R_0$ is the internal resistance of the MQ135 and it is required to know the concentration of gas because this value is obtained as the relationship between the internal resistance of the sensor $R_{Zero}$ and the measured resistance Rs [42]. Figure 9 shows the process of reading the value of this resistance. The process takes around half an hour before reaching a stable value (75.27 Ω). After calibrating the sensor, it will need approximately 20 s to heat and give accurate measures [43].

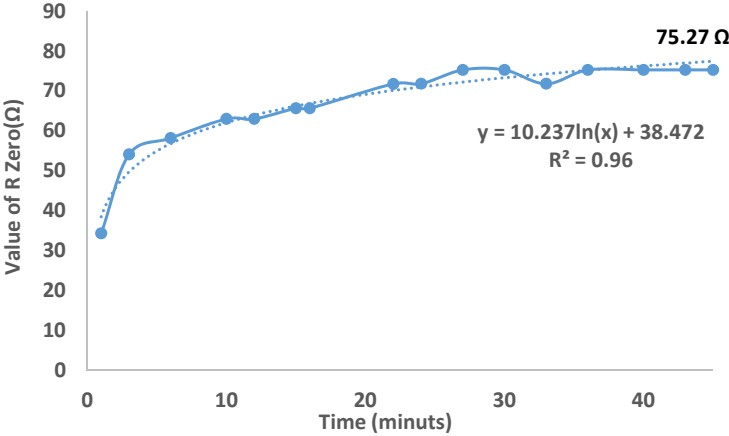

**Figure 9.** Measurement of R zero obtained after 45 min.

Finally, to measure the wind speed, a DC motor is used as a dynamo with a set of blades to capture wind and activate the motor. The blades have been designed as three linked hemispheres forming an equilateral triangle. The final result of our device is shown in Figure 10.

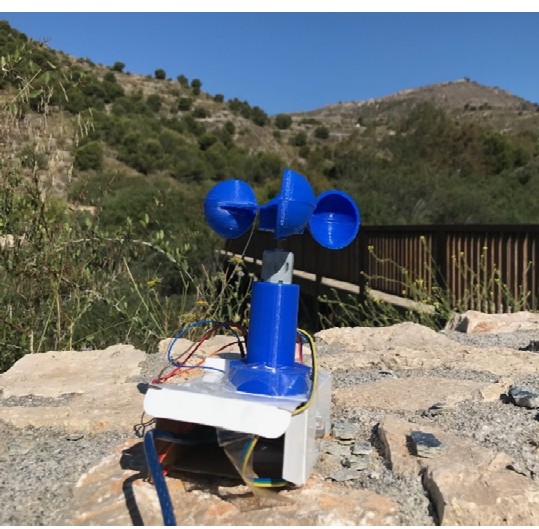

**Figure 10.** Node with the anemometer.

To measure the wind speed, a DC motor as a generator was used. The first step is to calibrate our device. To perform this task, a commercial anemometer is used and the output voltage of our DC motor is compared. Figure 11 shows the result of our calibration as well as the mathematical expression that models its behavior.

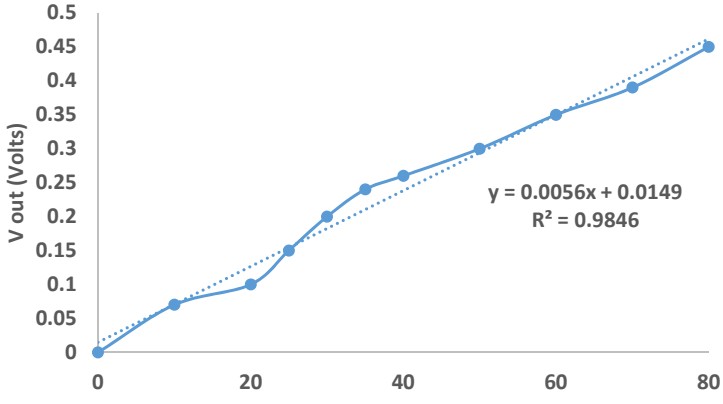

**Figure 11.** Calibration of anemometer.

Our purpose to determine if there is fire risk is to know when the wind speed is higher than 30 km/h. Because we also want to monitor the exact value of wind speed, the sensor was calibrated along with the entire measurement range.

*4.4. Gateway*

To connect the different nodes, a gateway manufactured by The Things Network is used [44]. This model is designed to be used indoors but, it could be used outdoors by using an adequate waterproof case. Table 2 shows the main features of this gateway.

**Table 2.** Features of TTN gateway.

| Parameter | Value |
|---|---|
| Channels | 8 |
| Working Frequency | 868 MHz (EU version) |
| Wireless Connectivity | LoRa and Wifi |
| Power | USB-C adapter with 900 mA |
| Antenna | Omnidirectional for indoor use |
| Manufacturer of the radio module | Semtech |

## 5. TTN Server Configuration and Application Server Design

As mentioned before, to implement our solution, the TTN platform will be used as a network server. This section shows the main steps to perform the correct configuration of platform.

To configure the server network, the TTN website will be accessed. The next step requires to create as many identifiers as nodes comprise the network. TTN will automatically assign the AppEUI key which will be used to connect the nodes to the network.

Once the application is created, the nodes will be registered as devices, and an identifier and a DevEUI of 8 bytes will be assigned. TTN will generate the AppKey later. Once it is created, the activation method of the device must be indicated. In this case, Activation By Personalization (ABP) is selected since it allows a faster link to the gateway. Finally, TTN will provide the session keys and the DevAddr that must be copied in the program code of each node. It is also possible to define the payload formats tab to visualize a decoder of the data sent by our nodes. With this, the user will be able to numerically see the data in the console, otherwise, it would display them in hexadecimal format.

### 5.1. Configuration of the Database (DB)

TTN allows configuring different integrations, one of them is a database that permits storing the node information in JSON format.

Within applications, an integration called data storage can be added. This integration has an API based on the REST representational state transfer system that allows requesting information to the database based on the cURL command interpreter. These requests will be HTTP requests to the database that should be identified with the key provided by TTN called Access Key.

### 5.2. Configuration of the Server Application

The application server will be an HTTP web server in charge of sending requests to the database implemented in the TTN server. It will also modify the data to transform it into a readable format. The web server will host an interpreter that will be executed continuously as a service of the operating system of the machine. This program is written in Python and it is mainly based on three functions:

- RequestHTTP(): This will be responsible for performing HTTP requests to the database, implemented on the TTN server. To do that, the system will use the Python module called "requests" which allows performing HTTP requests. The database will determine the time interval of the collected data by a parameter of the HTTP request, using the sequence of characters "1h". The response of the DB is kept in a variable that the JSON module will interpret as a vector of JSON objects. Finally, the last object, which correspond to the last measure performed by the node, will be kept.
- toParse(): This function is in charge of transforming the JSON data into a readable format to be interpreted by the program on the web (i.e., a format that can be drawn by the Google Maps API). It will save the JSON values for a determined node, being CO2 level, relative humidity value, temperature, wind speed, and the time stamp. We have a dictionary with the coordinates of each device (node) and another dictionary in a JSON format embedded in the argument of the "eqfeed_callback" function that is readable for the web application. It will save the values of the previous variables, identify the source node and, save the value of its coordinates in the dictionary. Finally, a file to save the data with *.json extension will be created.
- main(): It is the main function of the program and will be in charge of calling both the "RequestHTTP()" and "toParse()" functions.
- eqfeed_callback(): it creates a file where the coordinates of the nodes that exceed the values imposed by the 30-30-30 rule will be saved. It gets the values of the JSON objects from the "data.json" file. It defines a marker that indicates fire if the value of the $CO_2$ concentration exceeds the 700 ppm threshold and displays an alert by the screen. This function also defines the information windows that will be displayed by clicking on the markers and their format. Finally, the function is in charge of painting both the markers and the heat map, in case the defined thresholds have been exceeded.

Figure 12 shows the workflow diagram of our system. The diagram shows the workflow of our phyton application, which is in charge of checking rules taking into account the received data. Additionally, it calls other functions to, for example, draw the map.

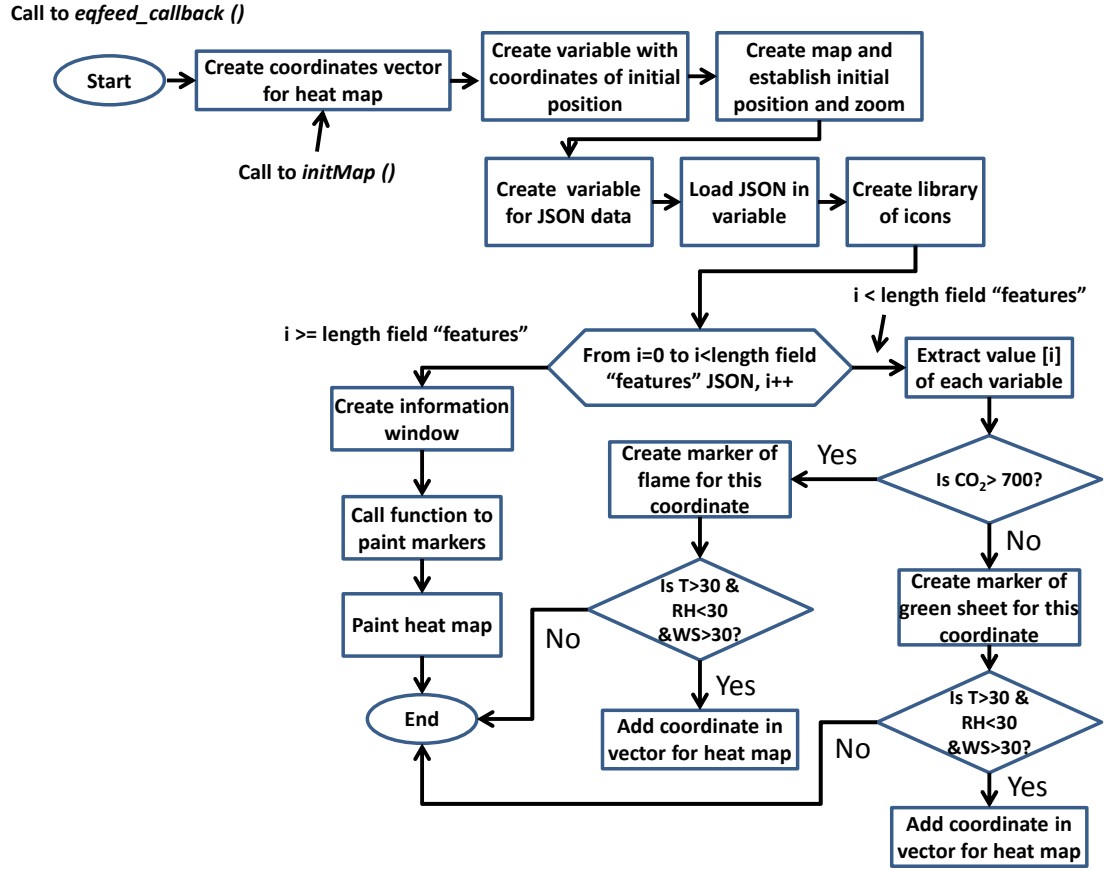

**Figure 12.** Workflow diagram of our system.

## 5.3. Web Interface

The web server will also host a website that implements a script written in the interpreted programming and object-oriented language of JavaScript. The script was made using the Google Maps API that allows working with the maps and satellite images that Google offers. In addition, it also permits the use of paint markers, information windows, polygons, heat maps, etc. Figure 13 shows the web interface developed to collect data from the nodes and visualize the values measured by them. The bottom part of the web site shows the legend with the icons used to represent different events, such as information of the nodes (a blue circle, ), the presence (a red flame, ) or no presence of fire (a green leaf, ), and the heat maps (a map, ). The results are updated every 5 min. In Figure 13, which shows a simulated case for graphical testing of the system, three green leaf icons are shown. This means that the $CO_2$ concentration is below the threshold. One of these icons is not evolved by the heat map of the zone. The heat areas are only shown when the values determined by the 30-30-30 rule have been exceeded. When the threshold of $CO_2$ is exceeded, i.e., there is fire, the icon of a flame will be displayed. When clicking on each icon, a new dialog box appears with the values of the measured environmental variables. Both the Presence of Fire and Detection of High Fire Risk are closely related since the presence of fire implies the level of $CO_2$ has values higher than 700 ppm while the combination of temperature sensor, RH sensor and wind speed with a value of $CO_2$ lower than 700 ppm will give us an alarm of High Fire Risk.

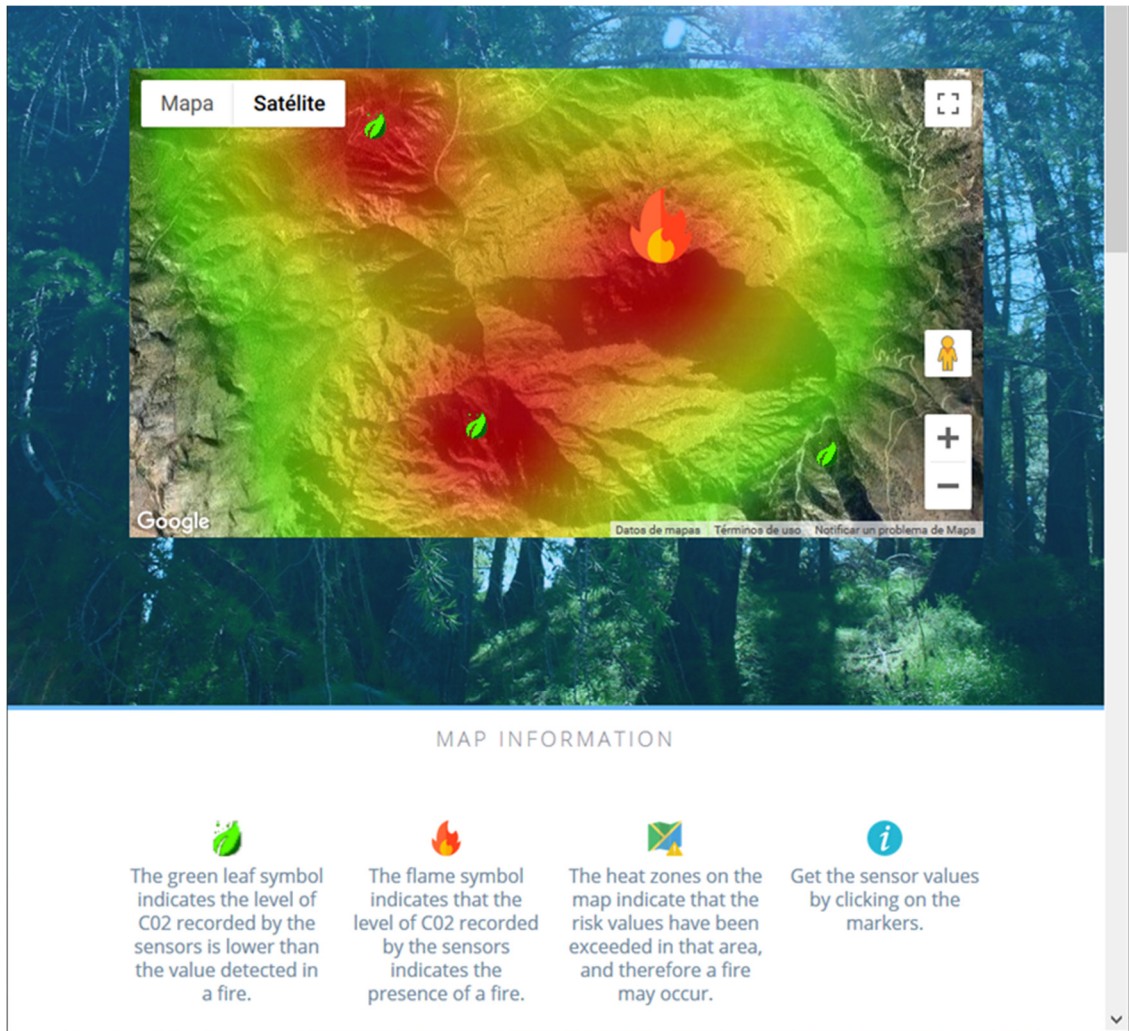

**Figure 13.** Website showing 4 nodes.

When the threshold is exceeded, a pop-up window (see Figure 14) is displayed on the screen. It indicates the coordinate of the node that detected the fire. By accepting the alert, the map is accessed to see the nodes and the measured values (see Figure 15).

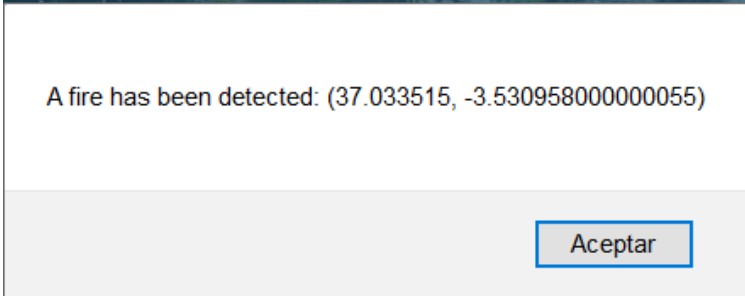

**Figure 14.** The Pop-up window when a fire is detected.

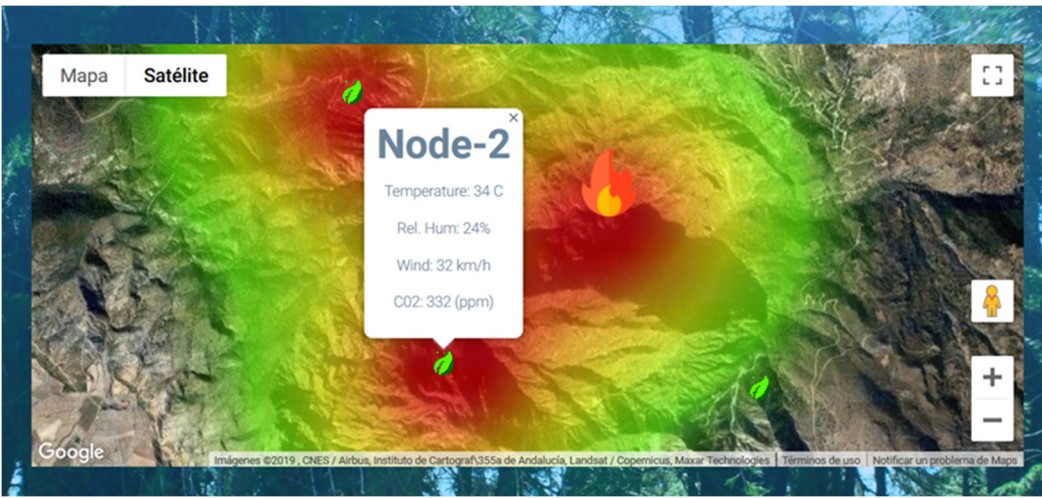

**Figure 15.** Website showing 4 nodes and the measured data.

The map shows the detected heat zones. The colored map identifies fire risk zones using green, yellow, orange, and red to indicate a low fire level, low-medium fire level, medium-high fire level, and high fire level, respectively. These are shown when the 30-30-30 rule values are exceeded. Figure 15 shows a scenario where three nodes detect values that exceed the thresholds of the 30-30-30 rule, where one of them has also detected a fire. There is a fourth node that does not detect dangerous levels.

## 6. Test Bench and Results

This section presents the test bench in terms of network coverage and the operation of the sensors in the entire system. The tests have been performed in a real environment.

### 6.1. LoRa Coverage

To perform the coverage test, a mixed scenario was chosen. It contains rural and urban areas. The scenario is located on the Guadalfeo river in the coastal town of Motril (Granada, Spain). The gateway is placed in an elevated position in the city of this scenario. SF = 7 was used because it is the configuration that would offer us the results in the worst possible scenario, i.e., SF7 offers the shortest coverage [24]. Figure 16 shows the position of the gateway.

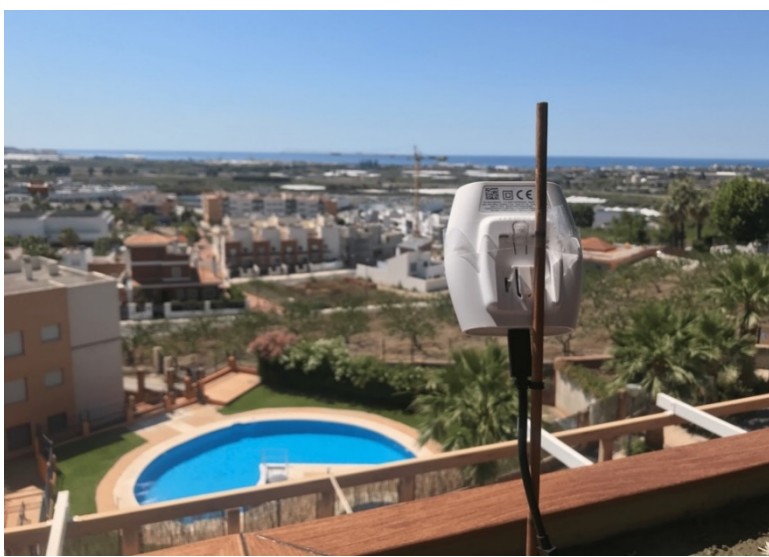

**Figure 16.** Position of the Gateway.

Using practical experiments, such as the ones shown in [45], we collected data of RSSI and SNR to determine the area that the network can cover and the quality of the received signal.

Figure 17 shows the RSSI values positioned on a real image of the scenario under study. The dots colored in red are the zones with the worst RSSI and, the green colored dots are the zones with a high RSSI. According to the results, we can see that a single gateway is giving coverage of 4km approximately.

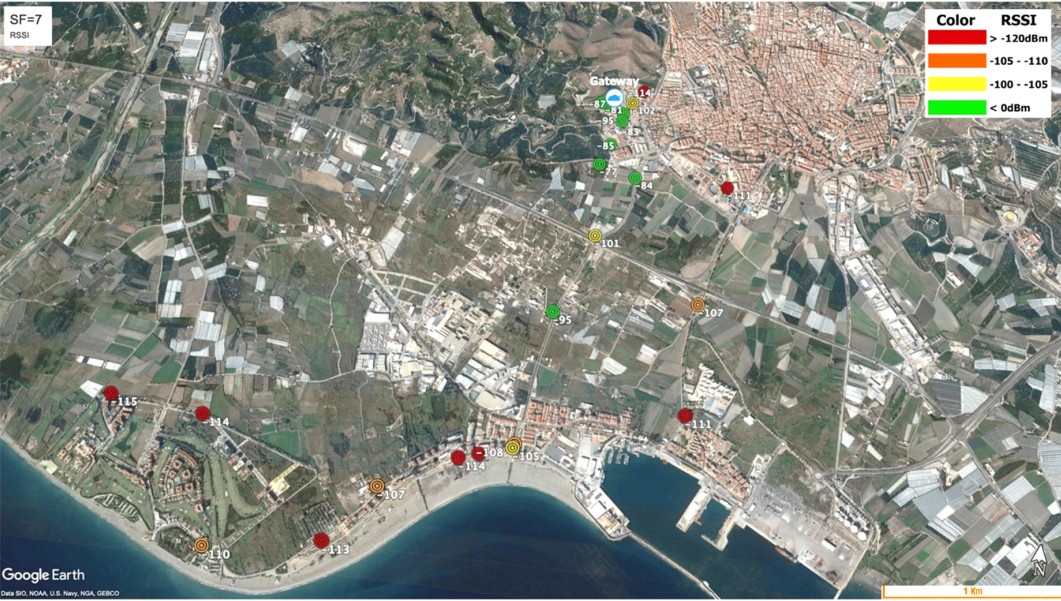

**Figure 17.** Results of RSSI measured in the scenario.

The RSSI values (see Figure 18) and SNR values (see Figure 19) are obtained based on the distance to the gateway. Values of received signal are compared to the Free Space Model [46,47]. The Free Space Model provides a measure of path-loss when the transmitter and receiver are within line of sight (LOS) range without any obstacles between them. It is based on the Friis free space transmission equation. We use this model to compare our results with the theoretical ones. As can be seen, the results follow the tendency of this ideal model although its values are worse.

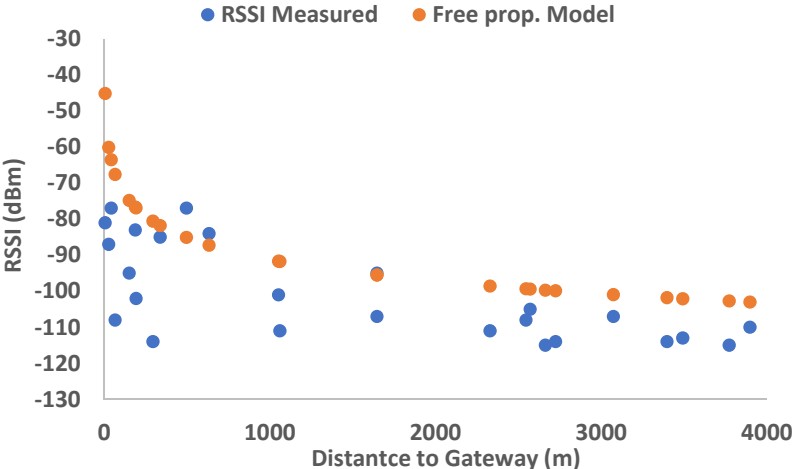

**Figure 18.** Results of RSSI as a function of the distance.

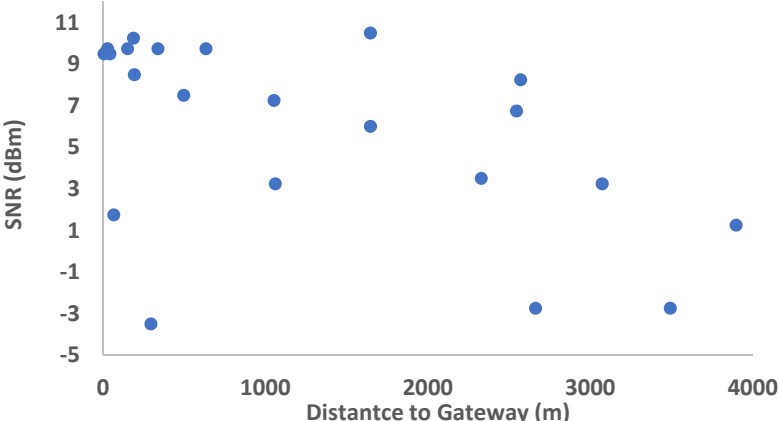

**Figure 19.** Results of SNR as a function of the distance.

We can observe that the coverage in the urban area is quite good when there is direct sight between the node and the gateway (i.e., in streets with few buildings in the Fresnel area between the devices). However, in the plain of the Vega, our coverage is very good, reaching values of −95 dBm at two kilometers away from the gateway. Near the coast, we can also observe the influence of the buildings, which reduce the obtained RSSI level. The maximum coverage range is obtained in the valley, with 4 km and an RSSI of −115 dBm.

Regarding the SNR values, they are very dispersed. The SNR can worsen due to interferences of similar signals (for example, working on the same frequency band or modulation) or due to a malfunction of wires or devices. The presence of extremely high moisture levels in the environment could worsen the signal as well, although this issue was not studied for LoRa networks. In this case, we are sure that no other LoRa node is transmitting in this area. If no interfering signal is present and we assume that devices are properly working, one could think that the SNR can be affected by the presence of obstacles in the line of sight. Therefore, this behavior could be due to the influence of buildings and obstacles.

### 6.2. Data from Sensors

To be sure of the correct operation of the node, the values measured by the sensors of a single node were checked for 20 h. The system was configured to take a measurement every 28 min.

Figure 20 shows the values of temperature measured during the test. We can see that the maximum temperature (23 °C) was registered at 22:40 approximately. This value decreases up to 25 °C at 4:15 a.m.

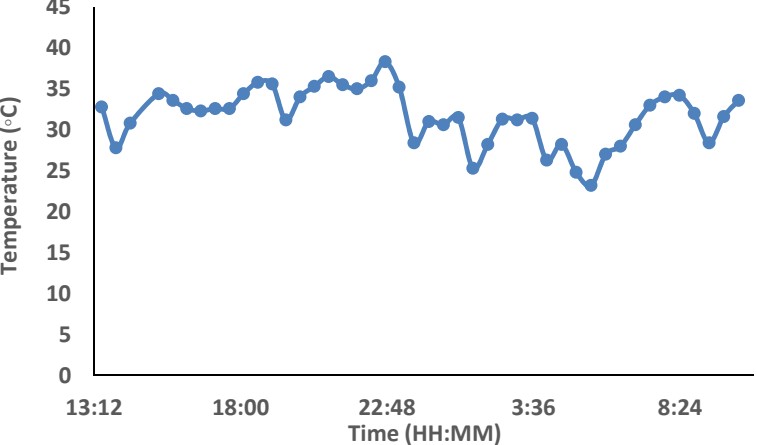

**Figure 20.** Temperature.

The relative humidity (see Figure 21) changes at daytime, i.e., the humidity increases from midnight due to the dew and air condensation that occurs in early mornings. It can be observed that the relative humidity is around 30% from 3:30 a.m. After the sunrise, in this case, the relative humidity remains with the same value. This is because the node being placed in a shady area.

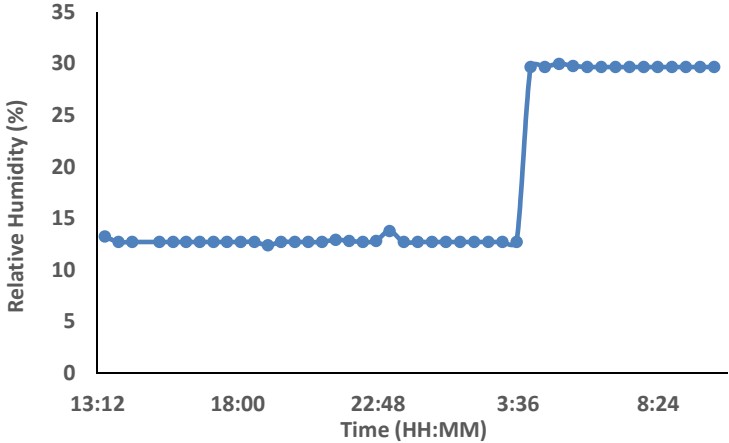

**Figure 21.** Relative Humidity.

Regarding wind speed (see Figure 22), we registered the maximum values from the evening to very early in the morning. Considering these values, one can determine that this is one of the worst scenarios when a fire is ongoing because emergency services cannot work at night.

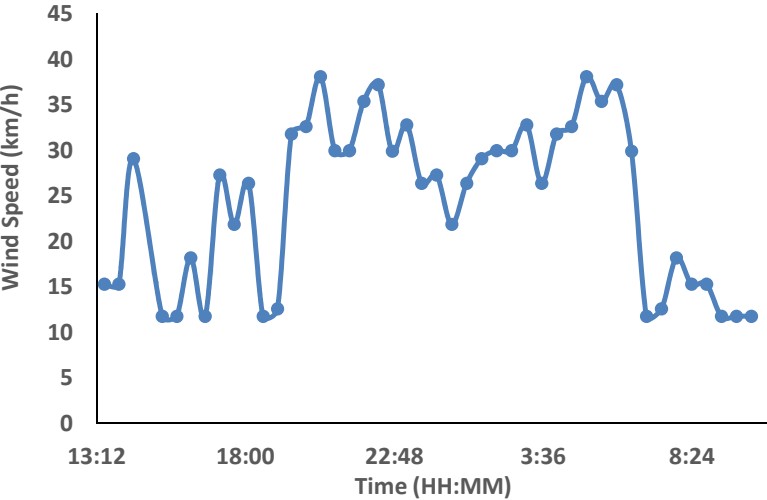

**Figure 22.** Wind speed.

Finally, Figure 23 shows the $CO_2$ concentration values recorded by the node. Since we could not generate a real fire, we emulated the presence of $CO_2$. To generate the smoke, we used a kitchen smoke pipe. As can be seen, the sensor was able to detect the presence of this gas which is the main gas that appears when a fire is detected.

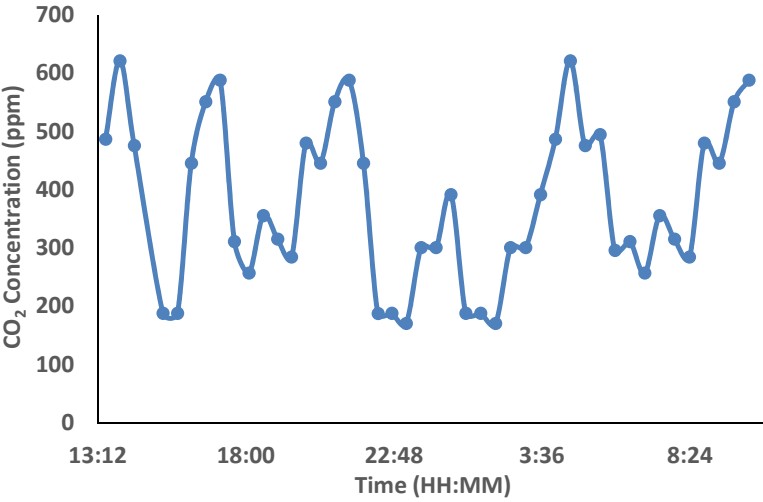

**Figure 23.** $CO_2$ Concentration.

Finally, we combine the data collected by this node to see how the alarms are generated (see Figure 24). In this case, we apply the algorithm shown in Figure 12. As we can see, the red line implements the 30-30-30 rule while the blue one detects the presence of $CO_2$. If the captured value of $CO_2$ surpasses the threshold of 620 ppm., the system will consider that there is a fire. These alarms will be translated into the correct symbol to be represented over the map and to generate the emerging message.

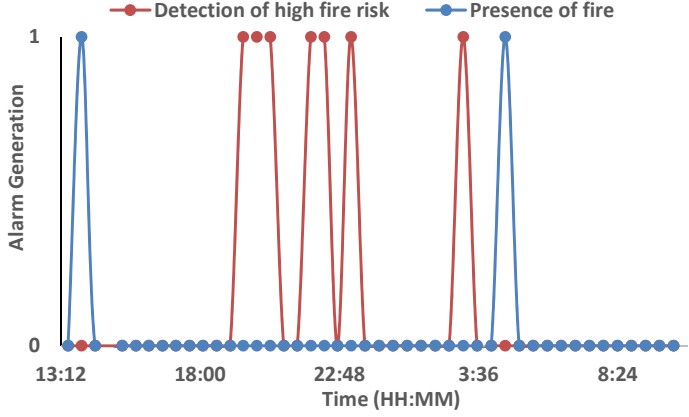

**Figure 24.** Alarms generated from the data collected by the node.

To finalize the test of our system, we estimate the power consumption of each node. The system is based on an Arduino mode that does not present the lowest value of energy consumption. However, it is possible to apply efficient programming to use sleep routines while no significant variations in the sensors are detected. Additionally, the RF transceiver can be maintained in this state for a long time and wake it up only to transmit a summary of the data or alarms. Alternative power sources such as solar energy can be used to power the node for several months.

The time our node can be powered by a battery can be estimated considering the consumptions of the devices specified by the datasheets. Firstly, we define three operating states for the wireless node to characterize each operation mode and to estimate the current consumed in each of these modes. The node will cyclically go through these three stages in the order that defines them. Subsequently, a schedule is defined to identify how much time the node will spend in each state.

- Stage 0. Preheating phase of MQ135: In this mode, the microcontroller is active and generates a call to the MQ135 sensor to start heating it. This process takes 20 s.

- Stage 1. Measurement of the sensors: In this mode, the microcontroller is active and generates a call to the sensors to obtain the parameters. In this mode, the microcontroller and communication peripherals are active, as well as the integrated circuits of the sensors.
- Stage 2. LoRa Transmission: The data is wirelessly transmitted by the node using the LoRa modulation. The microcontroller and the LoRa transceiver are transmitting at maximum power.
- Stage 3. Sleep mode: This will be the main stage of the node in which it will spend most of the time. In this stage, all modules are inactive using their sleep modes.

Table 3 shows a summary of the states for each module

**Table 3.** Stages of modules in each state.

| Stage | MCU | Sensors | LoRa Tx |
|---|---|---|---|
| Preheating phase of MQ135 | ON | Only MQ135 | OFF |
| Measurement of sensors | ON | ON | OFF |
| LoRa Tx | ON | OFF | ON |
| Sleep Mode | OFF | OFF | OFF |

We can estimate the current consumed by each module in each stage, according to the information collected in the data sheets (see Table 4).

**Table 4.** Consumption for each device.

| Device | Consumption in ON (mA) | Consumption in OFF (mA) |
|---|---|---|
| DHT11 | 2.5 | 0.15 |
| MQ135 | 150 | - |
| Arduino UNO | 46 | 0.058 |
| Dragino | 10.3 | 0.2 |

With this data, we add the consumptions in each operating mode according to the states indicated above (see Table 5).

**Table 5.** Consumption for each device.

| Status | Arduino UNO | Sensors | LoRa Tx | Total |
|---|---|---|---|---|
| Preheating phase of MQ135 | 46 mA | 150 mA | 0.2 mA | 196.2 mA |
| Measurement of sensors | 46 mA | 152.5 mA | 0.2 mA | 198.7 mA |
| LoRa Tx | 46 mA | 0.15 mA | 10.3 mA | 46.45 mA |
| Sleep Mode | 0.058 mA | 0.15 mA | 0.2 mA | 0.408 mA |

As can be seen, the mode that consumes more current is the state where the sensors are collecting the data from the environment, which is almost four times higher than the process of wirelessly transmitting the message with the information.

To estimate the amount of battery required to power a node for one year, the diagram of the operating time of the device in each state was obtained (see Figure 25). It should be noted that the schedule is not made to scale since the time spent in states 1 and 2 is very short compared to the sleep mode or state 3. As we commented before, the node sends data every 28 min. Therefore, 1,680,000 ms remain in an inactive state. The time remaining in state two, which is transmitting, correlates to the estimation of the air-time (107.78 ms) corresponding to a 55-byte payload with SF7 and 125 BW.

| Stage 0 | Stage 1 | Stage 2 | Stage 3 |
|---------|---------|---------|---------|
| 20,000ms | 1000ms | 107.78 ms | 1.680.000 ms. |
| 196.2 mA | 198.7 mA | 46.45 mA | 0.408 mA |

**Figure 25.** Diagram operating time of the device in each state.

Finally, it is possible to calculate the capacity (in mAh) the node needs in a cycle by using expressions (7)–(10):

$$196.2 \text{ mA} \times 20000\text{ms} = 3924000 \text{ mAms} \times \frac{1\text{h}}{3600000\text{ms}} = 1.09\text{mAh} \tag{6}$$

$$198.7 \text{ mA} \times 1000\text{ms} = 198700 \text{ mAms} \times \frac{1\text{h}}{3600000\text{ms}} = 0.552\text{mAh} \tag{7}$$

$$46.45 \text{ mA} \times 107.78\text{ms} = 5006,381 \text{ mAms} \times \frac{1\text{h}}{3600000\text{ms}} = 0.00139\text{mAh} \tag{8}$$

$$0.408 \text{ mA} \times 1680000\text{ms} = 685440 \text{ mAms} \times \frac{1\text{h}}{3600000\text{ms}} = 0.1904\text{mAh} \tag{9}$$

Table 6 shows a summary of these results.

**Table 6.** Summary of consumption calculation per cycle.

| Status | Capacity Required per Cycle |
|--------|-----------------------------|
| Preheating phase of MQ135 | 1.09 mAh |
| Measurement of sensors | 0.552 mAh |
| LoRa Tx | 0.00139 mAh |
| Sleep Mode | 0.1904 mAh |
| Total transmission | 1.833 mAh |

Therefore, to perform a full cycle, our node uses 1.833 mAh of energy. If we had a battery of 20,000 mAh of capacity, we could perform 10,911 transmissions, i.e., performing 52 transmissions per day (one transmission each 28-min intervals). Therefore, we could maintain the node active for 210 days (without taking into account the self-discharge. In this sense, the battery lifetime will be slightly lower).

Furthermore, the number of gateways we need to have full coverage of a wooded area can be defined. According to the results shown in Figure 18, considering the use of a gateway with an omnidirectional antenna [48], a single gateway is capable of covering a circular area of 4 km of radius. This implies that we can cover an area of approximately 50 km$^2$. We can, for example, consider some rural areas declared as protected natural parks in Spain and their total areas (see Table 7). Furthermore, to ensure that no place in the area is without connectivity, we can consider an overlapping of coverage of 30%. With these data, we can estimate the number of gateways necessary to cover the total area. Figure 26 shows the number of gateways required as a function of the area in ha. As can be seen, we could cover an area of 70,953 ha. using only 82 gateways.

Finally, it is defined a method for the Internet connection of the different gateways. Currently, some mobile operators provide services specifically designed for IoT and Machine-to-Machine (M2M) applications at a very low price. Therefore, each gateway will have a 3G dongle to permit the Internet connection and to send the data from the sensors to the server [47].

**Table 7.** Protected natural parks in Spain and their surface.

| Natural Parks in Spain | Surface (ha) | Surface (Km$^2$) | N° of Gateway |
|---|---|---|---|
| Garajonay | 3984 | 39.84 | 5 |
| Timanfaya | 5107 | 51.07 | 6 |
| Tablas de Damiel | 5410 | 54.1 | 6 |
| Caldera de Taburiente | 5956 | 59.56 | 7 |
| Doñana | 13,540 | 135.4 | 16 |
| Aigua Tortas y Lago de San Mauricio | 14,119 | 141.19 | 16 |
| Ordesa y Monte Perdido | 15,608 | 156.08 | 18 |
| Monfragüe | 17,852 | 178.52 | 21 |
| Teide | 18,900 | 189 | 22 |
| Sierra de Guadarrama | 33,960 | 339.6 | 39 |
| Cabañeros | 39,687 | 396.87 | 46 |
| Picos de Europa | 67,455 | 674.55 | 78 |
| Sierra Nevada | 70,953 | 709.53 | 82 |

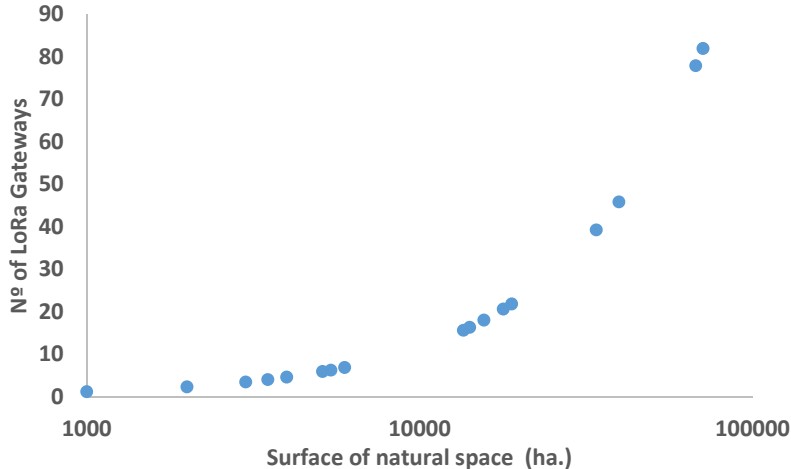

**Figure 26.** Required gateways as a function of the surface.

## 7. Conclusions

When fire burns natural landscapes, there is a huge impact on the ecosystem that negatively affects both the fauna and flora. Forest fires produce large amounts of greenhouse gasses such as $CO_2$ that worsen the problem of global warming and even affect citizens producing, in many cases, respiratory problems. Many of the fires that occur annually are caused by the carelessness of users or the lack of knowledge of those who burn stubble in an area with a high risk of fire. Therefore, the need for developing an autonomous system capable of assessing the risk indexes of a forested area from the measured parameters of the environment is evident.

This paper presented the design, development, and test of a low-cost system capable of determining the level of risk of suffering a fire in a rural area. The system is based on LoRa technology and is comprised of a set of nodes and a gateway. Each node has three sensors to measure the temperature, relative humidity, wind speed, and $CO_2$ of the environment to evaluate the level of fire risk and the presence of forest fires in an area. The data is sent to a network server in charge of processing and showing it graphically. The entire system was tested in a real environment by using an SF = 7. The scenario where the experiments were performed shows large plant masses that cause a large dispersion in the signal. However, we observed coverage ranges of 4 km which is a very interesting result considering a similar deployment based on WiFi.

Our web platform also implements a very easy to understand interface to consult the fire risk in an area. The platform takes the data from the sensors deployed in the environment. Furthermore, two

different types of alarms, fire presence, and fire risk can be generated as a function of those values and the operation algorithm of our system.

As future work, we would like to implement saving energy methods for our network in order to increase network stability and to reduce global consumption [49]. We want to combine the use of unmanned aerial vehicles for both environmental monitoring [50] and to implement a complementary system for emergency rescue services [51]. We want to validate our preliminary results and the algorithm in real situations.

Additionally, we assume that within the monitored area we will have to spread several nodes that will depend on factors such as the location and orography of the land. This would require an orographic study of the area to be monitored, so as to determine the variation of parameters such as temperature, humidity or even the condition of the field. With this, we could identify how significant a measure taken by a node is (i.e., determining the density of the nodes) and where we should place the next one for the results to be significant. The number of nodes does not have to be fixed since, if the nodes are in an area with great slope, the environmental parameters can vary rapidly and therefore more nodes would be required. However, if the nodes are in a flat area, the variations in the measurements are usually small and therefore, with a lower number of nodes, we could extract quite significant results. Therefore, in the near future, we would like to perform this study with specialized personnel.

Finally, considering the ratio of success of some pattern recognition methods that employ artificial intelligence techniques, we think the system would increase the efficiency in the estimations of fire risk by using these techniques.

**Author Contributions:** Conceptualization, S.S., J.L. and L.G.; methodology, L.G.; validation, R.V.-R. and I.B.; formal analysis, I.B. and R.V.-R.; writing—review and editing, S.S. and J.L. All authors have read and agreed to the published version of the manuscript.

**Funding:** This work was partially supported by the "Ministerio de Ciencia, Innovación y Universidades" through the "Ayudas para la adquisición de equipamiento científico-técnico, Subprograma estatal de infraestructuras de investigación y equipamiento científico-técnico (plan Estatal I+D+i 2017-2020)" (project EQC2018-004988-P), by Universidad de Granada through the "Programa de Proyectos de Investigación Precompetitivos para Jóvenes Investigadores. Modalidad A jóvenes Doctores" of "Plan Propio de Investigación y Transferencia 2019" (PPJIA2019.10), by the Campus de Excelencia Internacional Global del Mar (CEI·Mar) through the "Ayudas Proyectos Jóvenes Investigadores CEI·Mar 2019", (Project CEIJ-020), by the European Union through the ERANETMED (Euromediterranean Cooperation through ERANET joint activities and beyond) (Project ERANETMED3-227 SMARTWATIR).

**Acknowledgments:** This work was partially supported by the "Ministerio de Ciencia, Innovación y Universidades" through the "Ayudas para la adquisición de equipamiento científico-técnico, Subprograma estatal de infraestructuras de investigación y equipamiento científico-técnico (plan Estatal I+D+i 2017-2020)" (project EQC2018-004988-P), by Universidad de Granada through the "Programa de Proyectos de Investigación Precompetitivos para Jóvenes Investigadores. Modalidad A jóvenes Doctores" of "Plan Propio de Investigación y Transferencia 2019" (PPJIA2019.10), by the Campus de Excelencia Internacional Global del Mar (CEI·Mar) through the "Ayudas Proyectos Jóvenes Investigadores CEI·Mar 2019", (Project CEIJ-020), by the European Union through the ERANETMED (Euromediterranean Cooperation through ERANET joint activities and beyond) (Project ERANETMED3-227 SMARTWATIR).

**Conflicts of Interest:** The authors declare no conflict of interest. The funders had no role in the design of the study; in the collection, analyses, or interpretation of data; in the writing of the manuscript, or in the decision to publish the results.

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
