# Peer review of "LoRaWAN Network for Fire Monitoring in Rural Environments"

_electronics, doi:10.3390/electronics9030531_

Round 1

Reviewer 1 Report

The authors have not answered to my comments. In particular, it is not clear how one can deploy a LoRaWAN network in a forest: how to connect the Gateways to the network server?

Furthermore, the authors still confuse LoRa and LoRaWAN. The title refers to a LoRa network: there is not such a thing as a LoRa Network.

Author Response

Response for Reviewer 1:

Thanks for reviewing our paper. Thank you for your pertinent comments and detailed observations. These were really useful to improve the quality of our paper. Our responses are in line, starting with “Answer:”.

Comment: The authors have not answered to my comments. In particular, it is not clear how one can deploy a LoRaWAN network in a forest: how to connect the Gateways to the network server?

Answer : According to [A], to connect the Gateways to the network server, it is required to used a UDP-based protocol in charge of controlling the different flows (Downlinks, Uplinks and control messages) present in the network. Additionally, and due to the problems of security, the communication between network server and applications is commonly done by using encripted protocols with IDs for nodes and keys to encript the data. The following text has been added in page 7

“The Gateway has the function of collecting data from the different notes included in the network and connecting them to the rest of network. In this process, three different flows can be observed [A]:

Downlinks flow: the router receives downlink messages from upstream and waits for a gateway to be available near the device. Finally, it builds downlink options from the gateway options and transmits it. Uplinks flow: the router receives uplink messages from gateways and parses the uplink content for MAC payload and activations. Finally, it computes new downlink windows, finds a broker to transmit the packet to, and transmits the uplink. Status messages: the router uses these messages to track the the active gateways.

To generate all these messages and transmit from a gateway to TTN, two different protocols, i.e., the legacy Semtech UDP protocol and the Gateway Connector protocol, are required.

With The Semtech UDP protocol, uplinks, statuses and downlinks are exchanged in a pseudo-JSON format between the gateway and the network server. The UDP protocol does not provide authentication and there is no encryption available. For this reason, gateway messages can be intercepted during transport.

Using the Gateway connector protocol, it is possible to provide security to the system. Gateways are identified by an ID and by a key. Sending a message to a router requires to know the combination. With the gateway connector protocol, messages can be exchanged through network protocols such as MQTT/MQTTS or using gRPC (that supports TLS encryption natively), if hardware and software support it.

[A] https://microchipdeveloper.com/lora:lorawan-architecture”

Comment: Furthermore, the authors still confuse LoRa and LoRaWAN. The title refers to a LoRa network: there is not such a thing as a LoRa Network.

Answer : This issue has been revised.

Reviewer 2 Report

Overall appreciation: Very important and actual theme. The IoT topic can be considered a new approach to rural fire environment. However some considerations must be take into account, as mentioned in Particular Appreciation.

Particular appreciation:

Line 23: 30-30-30 rule should be more clear. Temperature/ Relative Humidity/Wind Speed: 30º - 30 RH - 30 km/h

Lines 49 and 50 should have a reference to the figures 1 and 2.

Line 82: The 30-30-30 rule must be explained. It is only explained in lines 346 to 348.

Line 177: The presented distances are based on other works or in this work?

Line 193: "expanded spectrum" should be replaced by "spread spectrum".

Line 194: LoRa has a lower bandwidth compared to other standards (125 kHz to 500 kHz)...

Line 199: 2 SF is wrong: is 2SF

Lines 200 and 201: for a SF = 7 the number of symbols is 128. Please explain the 27 chirps.

Line 204: Rs-> symbol/s .

Line 228: reference or explanation is needed.

Lines 229 to 233: the statements made in those paragraphs should have references. 

Line 288: KHz should be replaced for kHz.

Line 390: 3 minutes or 30 minutes?

Lines 405 and 406: This sentence does not make much sense, because the system must be calibrated to return the wind speed greater than 30km/h...

Line 408: the sentence is incomplete.

Table 2: some words are written in Spanish

Figure 14: GPS coordinate does not have such high resolution...

Line 494: Why is SF=10 the best value? Did this conclusion is based on other works or in previous measurements?

Figure 17: SF=7. The obtained RSSI values is for SF=7 or 10, as mentioned in line 494.

Figure 18: The obtained results is for a Line of Sight communication (LoS)? Maybe same end devices has no LoS with the gateway, therefore the values are much different than the expected ones.

Figure 18 and 19: there are some ate the same distance to the gateway. Is that correct?

Lines 524-531: explain the difference in figure 18...

Figures 20-23: What is the acquisition time between each sample? Values from figure 21 must be revised. HR duplicates within to samples... something is wrong...

Figure 24: There is any match between Presence of Fire and Detection of High Fire Risk. This feature must be explained.

Line 600-602: The SF=7 and BW=125kHz, however the Gateway is operating with SF=10 (line 494).

Lines 620-628: Figure 18 is RSSI vs distance. The gateway (antenna) is presented in Figure 16. However, that antenna is not omni-directional - the authors should consider present the antenna reference. How can the authors refer that can cover a circular area of 4 km radius? This paragraph should be revised. Due to this observation, revise the number of proposed gateways presented in Table 2.

Line 626 / Figure 26: How the number of gateways are obtained? There is a direct relation between the area and SF?

Line 656: SF=10, but in some parts of the paper it is referred SF=7.

Final considerations:

 The authors should validate, in future work, the preliminary results presented in this paper, namely the algorithm presented in Figure 12. Maybe the obtained results presented in Figure 24 will improve, such as when there is presence of fire the detection the risk must be activated.

 Other factor that must be take into account is the obtained values from the sensors, namely the RH sensor.

 The LoS and no - LoS communication must be take into account, in order to obtain valid RSSI values.

Author Response

Response for Reviewer 2:

Answer : Thanks for reviewing our paper. Thank you for your pertinent comments and detailed observations. These were really useful to improve the quality of our paper. Our responses are in line, starting with “Answer:”.

Comment: Line 23: 30-30-30 rule should be more clear. Temperature/ Relative Humidity/Wind Speed: 30º - 30 RH - 30 km/h

Answer : Another reviewer recommended not to add the concept of “30-30-30 rule” in the abstract. So, we have removed it from the abstract and the definition has been improved the first time it appears in lines 80-85. The following text has been added:

“This rule permits estimating the level of fire risk in an area taking into account the values of relative humidity, wind speed and temperature. A high fire risk  implies to have a temperature greater than 30º C, a relative humidity lower than 30%, wind speed greater than 30 km/h and absence of precipitation in the last 30 days.”

Comment: Lines 49 and 50 should have a reference to the figures 1 and 2.

Answer : Figures 1 y 2 have been done by using the data of reference [1] but they have been drawn by ourselve. However, we have added the reference in the sentence where we cite the figures:

“Figure 1 shows the number of wildfires in Europe, Middle East and North Africa in 2018 [1] while Figure 2 shows the burned hectares by wildfires in Europe, Middle East and North Africa in 2018 [1].”

Comment: Line 82: The 30-30-30 rule must be explained. It is only explained in lines 346 to 348.

Answer : The concept has been explained the first time it apperars in lines 80-85. The following text has been added:

“This rule permits estimating the level of fire risk in an area taking into account the values of relative humidity, wind speed and temperature. A high fire risk  implies to have a temperature greater than 30º C, a relative humidity lower than 30%, wind speed greater than 30 km/h and absence of precipitation in the last 30 days.”

Comment: Line 177: The presented distances are based on other works or in this work?

Answer : These values have been extracted from several papers with experimental results. Some of them are included in the related work section. We have also included the following references.

Sanchez-Iborra, R., Sanchez-Gomez, J., Ballesta-Viñas, J., Cano, M. D., & Skarmeta, A. F. (2018). Performance evaluation of LoRa considering scenario conditions. Sensors, 18(3), 772. Benites, B., Chávez, E., Medina, J., Vidal, R., & Chauca, M. (2019, February). LoRaWAN applied in Swarm Drones: A focus on the use of fog for the management of water resources in Lima-Peru. In Proceedings of the 5th International Conference on Mechatronics and Robotics Engineering (pp. 171-176). https://lora.readthedocs.io/en/latest/

Comment: Line 193: "expanded spectrum" should be replaced by "spread spectrum".

Answer : It has been corrected

Comment: Line 194: LoRa has a lower bandwidth compared to other standards (125 kHz to 500 kHz)...

Answer : Authors are not sure, whats the meaning of this comment since in this line there is no comment related to the bandwidth. Values commented in some lines before has been extracted from (https://lora.readthedocs.io/en/latest/)

The reference to this website has been added at the beginning of section 3.

Comment: Line 199: 2 SF is wrong: is 2SF

Answer : It has been corrected

Comment: Lines 200 and 201: for a SF = 7 the number of symbols is 128. Please explain the 27 chirps.

Answer : It has been corrected. It was an error the correct values is 2^7.

Comment: Line 204: Rs-> symbol/s.

Answer : It has been corrected

Comment: Line 228: reference or explanation is needed. Lines 229 to 233: the statements made in those paragraphs should have references. 

Answer : In some references stated tha SFs can be considered as ortogonal and for this reason we added the comment. However, and after Reading recent theoretical studies, We have seen that this statement cannot be true depending on the certain conditions. So SFs present some orthogonality imperfections they can interfere between each other. The following text (and references) have been added on page 6.:

“Regarding to the use of different channels, signals received with different SFs are normally considered purely orthogonal. However, it is not true under certain power level conditions. The end node parameters (SF and transmit power) can be adjusted based on the distance from the gateway and it offers the possibility to run networks with multiple gateways. [A][B][C]”

[A] Croce, D.; Gucciardo, M.; Tinnirello, I.; Garlisi, D.; Mangione, S. Impact of spreading factor imperfectorthogonality in lora communications.InInternational Tyrrhenian Workshop on Digital Communication;Springer: Cham, Switzerland, 2017; pp. 165–179

[B] Croce, D.; Gucciardo, M.; Mangione, S.; Santaromita, G.; Tinnirello, I.Impact of LoRa ImperfectOrthogonality: Analysis of Link-Level Performance.IEEE Commun. Lett.2018,22, 796–799. [CrossRef]

[C] Mikhaylov, K.; Petäjäjärvi, J.; Janhunen, J. On LoRaWAN scalability: Empirical evaluation of susceptibility tointer-network interference. In Proceedings of the 2017 European Conference on Networks and Communications(EuCNC), Oulu, Finland, 12–15 June 2017; pp. 1–

Comment: Line 288: KHz should be replaced for kHz.

Answer : It has been corrected

Comment: Line 390: 3 minutes or 30 minutes?

Answer : After calibrating the sensor, the device needs a preheat of 3 minuts to correctly performing the measurement. Before being able to measure, a calibration process that usually lasts around 24 hours is recommended. Rzero is obtained during the calibration process and it is obtained after the first 30 minutes.

Comment: Lines 405 and 406: This sentence does not make much sense, because the system must be calibrated to return the wind speed greater than 30km/h...

Answer : Maybe the sense of this sentence is not properly expressed. The idea is that our system needs to know when the wind speed is higher than 30km / h to determine if there is fire risk in a zone. But obviously, if we want to monitor the exact value of wind speed, the sensor should be calibrated to measure this parameter as accurate as possible.

The sentence has been rewritten as follow: Our purpose to determine if there is fire risk is to know when the wind speed is higher than 30km/h. Because we also want to monitor the exact value of wind speed, the sensor has been calibrated along the entire measurement range.

Comment: Line 408: the sentence is incomplete.

Answer : The sentence has been modified as follow:

“To connect the different nodes, a gateway manufactured by The Things Network is used [A]”

[A] The Things Indoor Gateway features. Available at: https://www.thethingsnetwork.org/docs/gateways/thethingsindoor/ [Last access: February 2, 2020].

Comment: Table 2: some words are written in Spanish

Answer : It has been corrected.

Comment: Figure 14: GPS coordinate does not have such high resolution...

Answer : We agree with the reviewer. The hardware used is probably simple, but it could be posible to develop the system with more accurate devices. However, the idea of having the coordinates is only to detect the position of node, not to tracking a mobile node.

Comment: Line 494: Why is SF=10 the best value? Did this conclusion is based on other works or in previous measurements?

Answer : Measurements were performed with SF=7; it was an error in the text.

Comment: Figure 17: SF=7. The obtained RSSI values is for SF=7 or 10, as mentioned in line 494.

Answer : Measurements were performed with SF=7; it was an error in the text.

Comment: Figure 18: The obtained results is for a Line of Sight communication (LoS)? Maybe same end devices has no LoS with the gateway, therefore the values are much different than the expected ones.

Answer : Our measurements have not been performed taking into account if nodes had LoS/n-LoS with the Gateway. The node were placed in several positions of  a real scenario and the values of snr and rssi were colected. We know that LoS/n-LoS positions give different results.

Comment: Figure 18 and 19: there are some ate the same distance to the gateway. Is that correct?

Answer : Yes, there are some measured points with very similar distances to the Gateway.

Comment: Lines 524-531: explain the difference in figure 18...

Answer : Our practical measurements (in blue) are compared with free-space model (in orange).The free-space model is a baseline model that provides a measure of path-loss when the transmitter and receiver are within line-of-sight (LOS) range without any obstacles between them. It is based on the Friis’ free-space transmission equation. The equation used to draw the orange line is:

PLFS(d)[dB] = 20 log10(f) + 20 log10(d) + 32.45

where d is the distance between the transmitter and the receiver in km, and f is the frequency in MHz.

The following text and references has been added after figure 17:

“The values of received signal is compared to the Free Space Model [A][B]. The Free Space Model provides a measure of path-loss when the transmitter and receiver are within line-of-sight (LOS) range without any obstacles between them. It is based on the Friis’ free-space transmission equation.”

[A] Linka, H., Rademacher, M., Aliu, O. G., & Jonas, K. (2018). Path loss models for low-power wide-area networks: Experimental results using lora. Available at: https://pdfs.semanticscholar.org/c362/dfd416bac03f2323204ec866baa0c00debae.pdf

[B] El Chall, R., Lahoud, S., & El Helou, M. (2019). LoRaWAN network: radio propagation models and performance evaluation in various environments in Lebanon. IEEE Internet of Things Journal, 6(2), 2366-2378.

Comment: Figures 20-23: What is the acquisition time between each sample? Values from figure 21 must be revised. HR duplicates within to samples... something is wrong...

Answer : The sampling time is 28 minutes. We have chequed and the number of samples for all graphs are the same.

Comment: Figure 24: There is any match between Presence of Fire and Detection of High Fire Risk. This feature must be explained.

Answer : Yes, the main diference is the level of CO2 Concentration, when there us fire the level of CO2 has values higher than 700 ppm while the combination of temperature sensor, RH sensor and wind speed with a value od CO2 lower than 700 ppm will give us an alarm of High Fire Risk.

The following thext has bee added before Figure 13: “Both the Presence of Fire and Detection of High Fire Risk are close related since the presence of fire implies the level of CO2 has values higher than 700 ppm while the combination of temperature sensor, RH sensor and wind speed with a value od CO2 lower than 700 ppm will give us an alarm of High Fire Risk.

Comment: Line 600-602: The SF=7 and BW=125kHz, however the Gateway is operating with SF=10 (line 494).

Answer : The SF used is 7. It was an error in the text (line 494)

Comment: Lines 620-628: Figure 18 is RSSI vs distance. The gateway (antenna) is presented in Figure 16. However, that antenna is not omni-directional - the authors should consider present the antenna reference. How can the authors refer that can cover a circular area of 4 km radius? This paragraph should be revised. Due to this observation, revise the number of proposed gateways presented in Table 2.

Answer : The number of gateways has been calculated taking into account an omnidirectional Gateway with the same power conditions of the one used in the test. For example, it is possible to use the (LoRa Lite Gateway). So taking into account an omni-directional antenna, the results are OK.

The following text  and the reference to the  LoRa Lite Gateway have been added before table 7:

According to the results shown in Figure 18, and considering the use of a gateway with an omni-directional antenna [], a single gateway is capable of covering a circular area of 4 km of radius.

[A] Lora Gateway Lite features. Available at: https://www.wireless-solutions.de/products/long-range-radio/lora-lite-gateway.html

Comment: Line 626 / Figure 26: How the number of gateways are obtained? There is a direct relation between the area and SF?

Answer : As reference [A] shows, the coverage area and the SF has a direct relation. When higher SF, higher coverage.

This reference has been added in subsection 6.1

[A] Waret, A., Kaneko, M., Guitton, A., & El Rachkidy, N. (2018). LoRa throughput analysis with imperfect spreading factor orthogonality. IEEE Wireless Communications Letters, 8(2), 408-411.

To estimate the number of Gateways, we consider the area of single Gateway and an overlaping of 25% between gateways (to be sure that the entire area is covered). After that, the total surface is divided into the coverage of a gateway (considering this overlaping).

Comment: Line 656: SF=10, but in some parts of the paper it is referred SF=7.

Answer : The SF used is 7. It was an error in the text (line 494)

Comment: The authors should validate, in future work, the preliminary results presented in this paper, namely the algorithm presented in Figure 12. Maybe the obtained results presented in Figure 24 will improve, such as when there is presence of fire the detection the risk must be activated.

Answer : Yes, we want to validate the algorithm as par of our future Works.

The following text has been added in the Future work paragraps (conclusión section): “We want to validate our preliminary results and the algorithm in real situations.”

Comment: Other factor that must be take into account is the obtained values from the sensors, namely the RH sensor.

Answer : The discuntinuity in Relative humity is due to the decreasing of wind speed. The tests wer performed near to the coast and the presence of dew is common. When the day is windy the dew decreases. When wind speed decreases the dew tends to increase. However 30% of RH is a very dry environment.

Comment: The LoS and no - LoS communication must be take into account, in order to obtain valid RSSI values.

Answer : Our measurements have not been performed taking into account if nodes had LoS/n-LoS with the Gateway. The node were placed in several positions of  a real scenario and the values of snr and rssi were colected. We know that LoS/n-LoS positions give different results. We will include this issue as future work.

Reviewer 3 Report

This paper presents the design and performance investigation of a LoRa sensor wireless network conceived for early fire detection in forestry. The authors have designed both the hardware and software portions of the network. The results are presented in terms of wireless coverage, which shows a good performance in an urban area. Moreover, an example of early fire detection is presented.

There are a number of issues with the current version of the paper, which in my opinion must be solved.

MAJOR

abstract, l. 23: delivering the "30-30-30" rule in the abstract is too early: the reader has to wait almost till to the end of the paper to discover its meaning. If it is not necessary in the abstract, please consider removing it from the abstract.

l. 80-81: a reference is needed to declare the superiority of LoRa with respect to SigFox in terms of data rate.

s. 2, related work: this section is quite comprehensive, but is very oriented towards wireless wildfire detection systems. Since LoRa is widely used also in other scenarios, from industry to agriculture to health, it could be interesting to add 2-3 references covering aspects other than wildfire detection and monitoring. The authors have a lot of recent works published in 2019 to choose from.

s. 3.1: the mathematical symbols in the text should be properly formatted in italic and with subscripts. For instance, l. 197 (Ts), l. 202 (Rs), l. 206 (Rc), and also SF and BW.  

l. 217, eq. 5: CR is both in the left side and in the denominator of the right side, there is a notation error probably.

l. 464-487: it is not clear if the web interface outlined in fig. 13 and described here is referred to a real case or if it is just a simulated case, for graphical testing of the system. Please clarify this point.

l. 468, fig. 13: it is stated that there are three green leaves in the figure, but I can only see two green leaves and one red leaf. Please also check the captions of fig. 13 and fig. 15 (3 or 4 nodes?).

l. 495: which is the "worst possible scenario" that justifies the adoption of the selected LoRa configuration?

l. 508: what do you consider as a "Free Propagation Model"? The free space loss model? Please add a reference or write and explain the equation used to draw the orange points in fig. 18.

l. 552: how did you emulate the presence of CO2?

l. 594: you write that the most power consuming state is that of the wireless transmission, however from tab. 5 it seems that the measurement state is the more onerous in terms of absorbed current. Please clarify and/or correct this point

l. 596-619, fig. 25: you write that the measurement state, with the largest current consumption of 150 mA, lasts for only 1 s. However, you previously wrote that the MQ135 needs at least 3 minutes (see l. 390) to have stable readings, after the initial 45 minutes calibration. In my experience, this type of sensor is very power consuming and also needs to be carefully managed in order to provide significant readings. It seems to me that the real battery capacity should be much larger, if an initial measurement time slot of 180 s (at least) is used. 

Minor

abstract/keywords: The Thing Network (l. 24) and The Think Networks (l. 29), please correct.

everywhere: what is the meaning of the yellow and blue highlighting of text? Results of previous revisions?

l. 37: "and, the amount" --> remove comma

l. 96: "Section6" --> missing space

l. 165: "stablished" --> perhaps "established"?

l. 199: "2SF" should be superscripted as 2SF

l. 201: "27" should be superscripted as 27 = 128

l. 204, eq. 1: "simbolos/s" --> "symbols/s"

l. 214: FEC means "forward error correction"

l. 220/223: what is the meaning of the bracketed (s)?

l. 236: "usually in which" --> "in which usually"

l. 261: can you add a reference for the usage of the term "speck"?

l. 265: "electricity network" --> "power grid"

l. 273: "it is regulated"

l. 287/288: define ToA before

l. 288: kilo is abbreviated with k, not with K. Also, it is kHz not kHs

l. 318, t. 1: "Price of end devices". Kbps or kbps? 2.4 instead of 2,4

l. 329: "along within the scenario"

l. 332: "and to forward"

l. 371: "form" --> "from"

l. 379: "analogical" --> "analog"

l. 382: "inside the sensor"

l. 391/402, fig. 9/10: what is the meaning and purpose of the equation written inside the figure? What does R2 stand for?

l. 393: what is "R zero"? Please explain.

l. 409: please add a reference for the used gateway.

l. 412, t. 2: "Omnidireccional etc." --> please translate in English

l. 421: what is ABP? Please explain.

l. 428: "our node information"

l. 475/480: perhaps the term "modal" is more appropriate than "emerging" or "emergent", please consider changing it

l. 499: "showed" --> "shown"

l. 505, fig. 17: maybe it would be clearer for the reader if a distance scale is added to the map, or maybe an arrow connecting the GW and the furthest operational node with the distance indicated. Please also consider explaining the correct interpretation of the legend shown on the map.

l. 534: which node has the results plotted in the figures? Please consider to indicate it.

l. 568: what is the "FR interface"?

l. 587/593, t. 3/5: "Tx LoRa" --> "LoRa Tx"

l. 590, t. 4: consumption is in mAh (charge) or in mA (current)?

l. 673: "in an area with great slope"

l. 684/689: the same text is used both for funding and acknowledgements sections

Author Response

Response for Reviewer 3:

Thanks for reviewing our paper. Thank you for your pertinent comments and detailed observations. These were really useful to improve the quality of our paper. Our responses are in line, starting with “Answer:”.

Comment: abstract, l. 23: delivering the "30-30-30" rule in the abstract is too early: the reader has to wait almost till to the end of the paper to discover its meaning. If it is not necessary in the abstract, please consider removing it from the abstract.

Answer : It has been removed from the abstact. The concept has been explained the first time it apperars in lines 80-85. The following text has been added:

“This rule permits estimating the level of fire risk in an area taking into account the values of relative humidity, wind speed and temperature. A high fire risk  implies to have a temperature greater than 30º C, a relative humidity lower than 30%, wind speed greater than 30 km/h and absence of precipitation in the last 30 days.”

Comment: l. 80-81: a reference is needed to declare the superiority of LoRa with respect to SigFox in terms of data rate.

 Answer : The following references have been added before table 1:

[A] A. Lavric, A. I. Petrariu and V. Popa, "Long Range SigFox Communication Protocol Scalability Analysis Under Large-Scale, High-Density Conditions," in IEEE Access, vol. 7, pp. 35816-35825, 2019.

[B] Finnegan, J., & Brown, S. (2018). A comparative survey of LPWA networking. arXiv preprint arXiv:1802.04222.

Comment: s. 2, related work: this section is quite comprehensive, but is very oriented towards wireless wildfire detection systems. Since LoRa is widely used also in other scenarios, from industry to agriculture to health, it could be interesting to add 2-3 references covering aspects other than wildfire detection and monitoring. The authors have a lot of recent works published in 2019 to choose from.

Answer : The following references have been added:

Froiz-Míguez, I.; Fraga-Lamas, P.; Varela-Barbeito, J.; Fernández-Caramés, T. M. LoRAWAN and Blockchain based Safety and Health Monitoring System for Industry 4.0 Operators. Proceedings 2019, 6, pp. 1-6. Doi: 10.3390/proceedings2019010006

Benites, B.; Chávez, E.; Medina, J.; Vidal, R.; Chauca, M. LoRaWAN applied in Swarm Drones: A focus on the use of fog for the management of water resources in Lima-Peru. In Proceedings of the 5th International Conference on Mechatronics and Robotics Engineering, Rome, Italy, 16-19 February 2019, pp. 171-176.

Sisinni, E.; Ferrari, P.; Fernandes Carvalho, D.; Rinaldi, S.; Pasetti, M.; Flammini, A.; Depari, A. A LoRaWAN range extender for Industrial IoT. IEEE Transactions on Industrial Informatics 2019, pp. 1-10. Doi: 10.1109/TII.2019.2958620

The following text has been added in the related work section:

“Iván Froiz-Míguez et al. presented in [A] a monitoring system for industrial workers to evaluate their safety and health in real-time. Sensors were deployed on IoT wearables that communicated with the closest gateway using LoRaWAN and LPWAN technology. The collected information was stored in distributed locations and processed using blockchain to provide immutability and traceability to the data when sharing it with medical facilities or insurance companies.

Boris Benites et al. proposed in [B] a water scarcity monitoring system intended for arid and semi-arid places. Humidity sensors and anemometers were incorporated to swarm drones to monitor zones with high concentrations of water. These drones forwarded the data using LoRaWAN to the registry for its storage. Then, the data would be transmitted to the central node to perform predict wind direction utilizing classification and regression techniques.

Emiliano Sisinni et al. designed in [C] a range extender for LoRaWAN industrial IoT. The authors suggested the use of a frame relay approach to avert compromising the highest data rates to higher sensitivity. The prototype was created from commercial hardware. Coverage tests were performed measuring the RSSI and the SNR to ensure the correct performance of the proposal and to demonstrate its effectiveness in increasing the range. Furthermore, the range extender was able to perform with legacy networks of LoRaWAN”

Comment: s. 3.1: the mathematical symbols in the text should be properly formatted in italic and with subscripts. For instance, l. 197 (Ts), l. 202 (Rs), l. 206 (Rc), and also SF and BW.  

Answer : It has been corrected.

Comment: l. 217, eq. 5: CR is both in the left side and in the denominator of the right side, there is a notation error probably.

Answer : It has been corrected. The correct expression is

Comment: l. 464-487: it is not clear if the web interface outlined in fig. 13 and described here is referred to a real case or if it is just a simulated case, for graphical testing of the system. Please clarify this point.

Answer : Picture shows a a simulated case for graphical testing of the system.

We have clarify this issue in the text, adding the following sentence (in subsection5.3):

“In Figure 13, which shows a a simulated case for graphical testing of the system”

Comment: l. 468, fig. 13: it is stated that there are three green leaves in the figure, but I can only see two green leaves and one red leaf. Please also check the captions of fig. 13 and fig. 15 (3 or 4 nodes?).

Answer : There are 3 green leaves and a flame, i.e., 4 nodes (see the circles). So, the figures captations are ok.

Comment: l. 495: which is the "worst possible scenario" that justifies the adoption of the selected LoRa configuration?

Answer : The SF 7 is the one that offers the worst results in terms of coverage. The following reference shows how the distance increases with the SF.

The following text has been added before figure 16: “…., i.e., SF7 offers the shortest coverage [A].”

[A] Waret, A., Kaneko, M., Guitton, A., & El Rachkidy, N. (2018). LoRa throughput analysis with imperfect spreading factor orthogonality. IEEE Wireless Communications Letters, 8(2), 408-411.

Comment: l. 508: what do you consider as a "Free Propagation Model"? The free space loss model? Please add a reference or write and explain the equation used to draw the orange points in fig. 18.

Answer : The free-space model is a baseline model that provides a measure of path-loss when the transmitter and receiver are within line-of-sight (LOS) range without any obstacles between them. It is based on the Friis’ free-space transmission equation. The equation used to draw the orange line is:

PLFS(d)[dB] = 20 log10(f) + 20 log10(d) + 32.45

where d is the distance between the transmitter and the receiver in km, and f is the frequency in MHz.

The following text and references has been added after figure 17:

“The values of received signal is compared to the Free Space Model [A][B]. The Free Space Model provides a measure of path-loss when the transmitter and receiver are within line-of-sight (LOS) range without any obstacles between them. It is based on the Friis’ free-space transmission equation.”

[A] Linka, H., Rademacher, M., Aliu, O. G., & Jonas, K. (2018). Path loss models for low-power wide-area networks: Experimental results using lora. Available at: https://pdfs.semanticscholar.org/c362/dfd416bac03f2323204ec866baa0c00debae.pdf

[B] El Chall, R., Lahoud, S., & El Helou, M. (2019). LoRaWAN network: radio propagation models and performance evaluation in various environments in Lebanon. IEEE Internet of Things Journal, 6(2), 2366-2378.

Comment: l. 552: how did you emulate the presence of CO2?

Answer : We used a kitchen smoke pipe that burns small pieces of wood and generates smoke. The tests with sensors were carried out in a natural park so no fire or similar were permited

The following text has been added before figure 23: “ To generate de smoke, we have used a kitchen smoke pipe.”

Comment: l. 594: you write that the most power consuming state is that of the wireless transmission, however from tab. 5 it seems that the measurement state is the more onerous in terms of absorbed current. Please clarify and/or correct this point

Answer : Yes, it is true, it was a mistake in the writting. The comment has been revised as follow:

“As it can be seen, the mode that consumes more current is the state where the sensors are collecting the data from the environment which is almost four times higher than the process of wirelessly transmitting the message with the information.”

Comment: l. 596-619, fig. 25: you write that the measurement state, with the largest current consumption of 150 mA, lasts for only 1 s. However, you previously wrote that the MQ135 needs at least 3 minutes (see l. 390) to have stable readings, after the initial 45 minutes calibration. In my experience, this type of sensor is very power consuming and also needs to be carefully managed in order to provide significant readings. It seems to me that the real battery capacity should be much larger, if an initial measurement time slot of 180 s (at least) is used.

Answer : We have recalculated the consumption, considering the preheating phase that, acording to this link is only 20 seconds. The battery lifetime has been done considering 20 Ah.

Figure 25. Diagram operating time of the device in each state.

                              (7)

Therefore, we obtain that to perform a complete cycle, our node uses 1.833 mAh of energy. If we had a battery of 20,000 mAh of capacity, we could perform 10,911 transmissions, i.e., performing 52 transmissions per day (one transmission each 28-minute intervals). Therefore, we could maintain the node active for 210 days (without taking into account the self-discharge. In this sense, the battery lifetime will be slightly lower).

Comment: abstract/keywords: The Thing Network (l. 24) and The Think Networks (l. 29), please correct.

Answer : It has been corrected. The correct is The things Network

Comment: everywhere: what is the meaning of the yellow and blue highlighting of text? Results of previous revisions?

Answer : In an earlier review phase, the reviewers recommended a series of changes. These changes were indicated in yellow, according to the Journals’ indications

Comment: l. 37: "and, the amount" --> remove comma

Answer : It has been corrected.

Comment: l. 96: "Section6" --> missing space

Answer : It has been corrected.

Comment: l. 165: "stablished" --> perhaps "established"?

Answer : It has been corrected.

Comment: l. 199: "2SF" should be superscripted as 2SF

Answer : It has been corrected.

Comment: l. 201: "27" should be superscripted as 27 = 128

It has been corrected.

Comment: l. 204, eq. 1: "simbolos/s" --> "symbols/s"

Answer : It has been corrected.

Comment: l. 214: FEC means "forward error correction"

Answer : It has been corrected.

Comment: l. 220/223: what is the meaning of the bracketed (s)?

Answer : The letter in parentheses is seconds. We have added:

“the duration (in seconds) can be calculated as”

Comment: l. 236: "usually in which" --> "in which usually"

Answer : It has been corrected.

Comment: l. 261: can you add a reference for the usage of the term "speck"?

Answer : It is a wrong word, the correct term is “mote”. It has been corrected

Comment: l. 265: "electricity network" --> "power grid"

Answer : It has been corrected.

Comment: l. 273: "it is regulated"

Answer : It has been corrected.

Comment: l. 287/288: define ToA before

Answer : This concept is defined the first time it appears (in page 6, after equation 6)

Comment: l. 288: kilo is abbreviated with k, not with K. Also, it is kHz not kHs

Answer : It has been corrected.

Comment: l. 318, t. 1: "Price of end devices". Kbps or kbps? 2.4 instead of 2,4

Answer : It has been corrected.

Comment: l. 329: "along within the scenario"

Answer : It has been corrected.

Comment: l. 332: "and to forward"

Answer : It has been corrected.

Comment: l. 371: "form" --> "from"

Answer : It has been corrected.

Comment: l. 379: "analogical" --> "analog"

Answer : It has been corrected.

Comment: l. 382: "inside the sensor"

Answer : It has been corrected.

Comment: l. 391/402, fig. 9/10: what is the meaning and purpose of the equation written inside the figure? What does R2 stand for?

Answer : (Fig 10) The wind sensor calibration equation allows us determining the equivalence between the voltage level measured by the arduino module and the actual wind speed in the environment.

Fig 9 simply shows how the value of the resistance varies over time until it reaches a stable value and that its behavior is logarithmic. R2 shows the similarity of equation and the behaviour of measurements

Comment: l. 393: what is "R zero"? Please explain.

Answer : R zero or R0 is the internal resistance of sensor. To ensure a correct measuremnt, this value should be measured as show in figure 9.

The following sentence has been added before figure 9:

“RZero or R0 is the internal resistance of the MQ135 and it is required to know the concentration of a gas because this values is obtained as the relationship between the internal resistance of the sensor RZero and the measured resistance Rs [29].”

Comment: l. 409: please add a reference for the used gateway.

Answer : The following senstence and reference has been added in subsection 4.4:

“To connect the different nodes we have used a gateway manufactured by The Things Network [A]”

[A] The Things Indoor Gateway features. Available at: https://www.thethingsnetwork.org/docs/gateways/thethingsindoor/ [Last access: February 2, 2020].

Comment: l. 412, t. 2: "Omnidireccional etc." --> please translate in English

Answer : It has been corrected.

Comment: l. 421: what is ABP? Please explain.

Answer : ABP is a mode of authentication of the nodes in the network. The acronym definition has been added on page 13 (the first time it appears) “Activation By Personalization (ABP)”

Comment: l. 428: "our node information"

Answer : It has been corrected.

Comment: l. 475/480: perhaps the term "modal" is more appropriate than "emerging" or "emergent", please consider changing it

Answer : we have changed “emerging window” by pop-up window

Comment: l. 499: "showed" --> "shown"

Answer : It has been corrected.

Comment: l. 505, fig. 17: maybe it would be clearer for the reader if a distance scale is added to the map, or maybe an arrow connecting the GW and the furthest operational node with the distance indicated. Please also consider explaining the correct interpretation of the legend shown on the map.

Answer : The distance scale is already placed in the map (see the orange circle).

However, we have edited the image to give more contrast and to be able od seeing the scale.

Comment: l. 534: which node has the results plotted in the figures? Please consider to indicate it.

Answer : Figure 13 shows a simulated case for graphical testing of the system. Graphs with the sensors results have been obtained placing the node in a point in the mountain for 20 hours but its position does not correspond with any place shown in Figure 13.

Comment: l. 568: what is the "FR interface"?

Answer : RF transceiver. It was wrongly written

Comment: l. 587/593, t. 3/5: "Tx LoRa" --> "LoRa Tx"

Answer : It has been corrected.

Comment: l. 590, t. 4: consumption is in mAh (charge) or in mA (current)?

Answer : Consumption is expressed in mAh. Table 4 has been corrected.

Comment: l. 673: "in an area with great slope"

Answer : It has been corrected.

Comment: l. 684/689: the same text is used both for funding and acknowledgements sections

Answer : The article is part of the aforementioned projects and they are also the ones that provide financing for the purchase of the devices and carry out the experiments. For that reason both sections include the same textslts. We will include this issue as future work.

Reviewer 4 Report

The paper presents the design, implementation and testing of a wireless sensor network based on LoRa protocol for fire risk assessment and early warning. Both underlying idea and implementation are described in detail, even though English usage seems to be poor and writing style sometimes superficial. Results of both nodes and platform are reported in a confused way, thus limiting the understanding of the potential efficacy of the proposed solution. This way, I guess that the paper cannot be accepted for publication in its current version.

In particular, the main concern is related to the novelty and originality of the paper. More specifically, the authors in Section 2 provide a rich list of technical solutions already presented in the literature for fire risk assessment; however, the list is not critically analyzed, highlighting limitations and/or drawback that the approach proposed by the authors should overcome. Reading the paper, you can think “Yet another fire-preventing network”. The authors are invited to accurately claim how their approach differ from those presented in the literature and why would be better; this is a fundamental task to be accomplished in order to make the paper suitable for publication.

As stated before, writing style appear to be superficial. Several typos are spread throughout the paper and some sentences have to be simplified or rewritten (e.g. lines 50-54 in the introduction). Spanish language sometimes crops out (e.g. “simbolos” in Eq.1 or “Omnidireccional para uso en interior” in Tab.2). In other words, authors are invited to pay more attention at the presentation.

In the following, a list of other concerns both minor and major

Consider avoiding the first person in the presentation (we make, we propose…) Parameters and quantities should be formatted in italic font Please check Eq.5 SF are orthogonal if the same BW is taken into account; there are some combinations of SF and BW characterized by the same slope of the frequency sweep and, consequently, is more correct speaking about pseudo-orthogonality Due to the stationarity of the proposed network, advantage of LoRa in terms of immunity to Doppler effect (presented in line 231-233) turns out to be useless I’m not sure that final device can be used with the same sense of end device Line 288, the closing parenthesis is missing Line 288 (and somewhere else, e.g. labels of axes of fig. 9 and 11) pay attention to symbols of measurement units; in this line is reported KHs instead of kHz I suggest switching the order of presentation in Section 3; more specifically, you have to first justify the selection of LoRa/LoRaWAN with respect to the other available solutions (as in Lines 297-307) and successively provide details about the chosen protocol Define acronym once (TTN is defined also in the abstract); for ABO no definition is given According to what reported in lines 349-352, using 2 parameters instead of 3 provides higher confidence levels in detecting fire-risk conditions; why do the authors exploit three parameters? According to current international recommendations, please provide the uncertainty rather than the error in line 377 11 presents the calibration curve of the DC motor for wind speed measurements; in order to appreciate its significance, the authors are invited to provide the repeatability (in terms of once or twice the standard deviation of repeated measurements) associated with experimental point. Moreover, the same calibration should be carried out on other DC motors to assess the reproducibility of the calibration curve 12 shows the workflow of the proposed system. Is the diagram associated with the Python application? Are the blocks related with the function implemented in Python? What is eqfeed_callback()? In the conditions “i>=length …” i should be lower case. In some blocks is written coordenate instead of coordinates. According to what stated in Line 495 and in the conclusion section, experimental tests have been carried out with LoRa signals characterized by SF equal to 10; on the contrary, reported results and consumption considerations are associated with SF=7 (e.g. Fig.17 and line 602). Description of results about temperature tests does not correspond to what shown in Fig.20 Why is there a discontinuity in the relative humidity curve? How do the authors emulate fire in tests with CO2 sensor? Also at night? Line 568 FR should be RF? Table 4 Consumptions of device should be in mA, not mAh According to what states Tab. 5, the most consuming operation is the measurement of sensor and not Tx LoRa as reported in the following lines 594-595 Line 600 It is better to write 1680000 ms Calculations for the definitions of consumptions of Tx LoRa have to be updated with the correct SF; if this is the case, consumption should be 8 times higher. In Eq. 8 e 9, the same consumption of 198700mAms is given; I think this is a “copy-and-past” error The authors do not taken into account self-discharge in determining the lifetime of the battery

Author Response

Response for Reviewer 4:

Thanks for reviewing our paper. Thank you for your pertinent comments and detailed observations. These were really useful to improve the quality of our paper. Our responses are in line, starting with “Answer:”.

Comment: In particular, the main concern is related to the novelty and originality of the paper. More specifically, the authors in Section 2 provide a rich list of technical solutions already presented in the literature for fire risk assessment; however, the list is not critically analyzed, highlighting limitations and/or drawback that the approach proposed by the authors should overcome. Reading the paper, you can think “Yet another fire-preventing network”. The authors are invited to accurately claim how their approach differ from those presented in the literature and why would be better; this is a fundamental task to be accomplished in order to make the paper suitable for publication.

Answer : Our paper has as a novelty, compared to already published papers, that it implements a fire detection system based on LoRa technology that uses the 30-30-30 rule. In addition, the current fire detection systems only detect the presence of fire. Our system is able to assess the fire risk levels of an area.

The following text has been added at the end of related work section: “Additionally, the system uses the 30-30-30 rule to determine if there is high level of fire risk. So our system is able to both predict and detect fires. Finally, “

Comment: As stated before, writing style appear to be superficial. Several typos are spread throughout the paper and some sentences have to be simplified or rewritten (e.g. lines 50-54 in the introduction). Spanish language sometimes crops out (e.g. “simbolos” in Eq.1 or “Omnidireccional para uso en interior” in Tab.2). In other words, authors are invited to pay more attention at the presentation.

Answer : It has been corrected.

Comment: Consider avoiding the first person in the presentation (we make, we propose…)

Answer : We have corrected several of them.

Comment: Parameters and quantities should be formatted in italic font

Answer : It has been corrected.

Comment: Please check Eq.5

Answer : It has been corrected.

Comment: SF are orthogonal if the same BW is taken into account; there are some combinations of SF and BW characterized by the same slope of the frequency sweep and, consequently, is more correct speaking about pseudo-orthogonality

Answer : In some references stated tha SFs can be considered as ortogonal and for this reason we added the comment. However, and after Reading recent theoretical studies, We have seen that this statement cannot be true depending on the certain conditions. So SFs present some orthogonality imperfections they can interfere between each other. The following text (and references) have been added on page 6:

“Regarding to the use of different channels, signals received with different SFs are normally considered purely orthogonal. However, it is not true under certain power level conditions. The end node parameters (SF and transmit power) can be adjusted based on the distance from the gateway and it offers the possibility to run networks with multiple gateways. [A][B][C]”

[A] Croce, D.; Gucciardo, M.; Tinnirello, I.; Garlisi, D.; Mangione, S. Impact of spreading factor imperfectorthogonality in lora communications.InInternational Tyrrhenian Workshop on Digital Communication;Springer: Cham, Switzerland, 2017; pp. 165–179

[B] Croce, D.; Gucciardo, M.; Mangione, S.; Santaromita, G.; Tinnirello, I.Impact of LoRa ImperfectOrthogonality: Analysis of Link-Level Performance.IEEE Commun. Lett.2018,22, 796–799. [CrossRef]

[C] Mikhaylov, K.; Petäjäjärvi, J.; Janhunen, J. On LoRaWAN scalability: Empirical evaluation of susceptibility tointer-network interference. In Proceedings of the 2017 European Conference on Networks and Communications(EuCNC), Oulu, Finland, 12–15 June 2017; pp. 1–

Comment: Due to the stationarity of the proposed network, advantage of LoRa in terms of immunity to Doppler effect (presented in line 231-233) turns out to be useless I’m not sure that final device can be used with the same sense of end device

Answer : The bibliography consulted (in some cases) talks indistinctly of the final device and final device, but to avoid any ambiguity, we have decided to express this element with the name (end device), as they are cited on the web of The Things network.

Comment: Line 288, the closing parenthesis is missing

Answer : It has been corrected.

Comment: Line 288 (and somewhere else, e.g. labels of axes of fig. 9 and 11) pay attention to symbols of measurement units; in this line is reported KHs instead of kHz

Answer : It has been corrected.

Comment: I suggest switching the order of presentation in Section 3; more specifically, you have to first justify the selection of LoRa/LoRaWAN with respect to the other available solutions (as in Lines 297-307) and successively provide details about the chosen protocol.

Answer : In a previous review, a review recommended simply adding a summary table of the technologies and references to LPWAN technologies. For this reason, the table is included at the end of the section. The objective of table 1 and figure 5 was to show the characteristics of LoRa technology with other similar technologies.

Comment: Define acronym once (TTN is defined also in the abstract); for ABO no definition is given

Answer : It has been corrected.

ABP is a mode of authentication of the nodes in the network. The acronym definition has been added on page 13 (the first time it appears) “Activation By Personalization (ABP)”

Comment: According to what reported in lines 349-352, using 2 parameters instead of 3 provides higher confidence levels in detecting fire-risk conditions; why do the authors exploit three parameters?

Answer : The study refers to use these parameters combining with the  rain. In our case we do not use rain but we included CO2 as a method of verifying the presence of a fire.

The following text has been added in subsection 4.2: “Since the rain is not used in this study, the CO2 concentration has been added as a method for verifying the presence of a fire.”

Comment: According to current international recommendations, please provide the uncertainty rather than the error in line 377. 11 presents the calibration curve of the DC motor for wind speed measurements; in order to appreciate its significance, the authors are invited to provide the repeatability (in terms of once or twice the standard deviation of repeated measurements) associated with experimental point. Moreover, the same calibration should be carried out on other DC motors to assess the reproducibility of the calibration curve

Answer : These data were directly exptracted from the datasheet and the manufacturers only provided the error in %.

Regarding to the wind speed sensors, we only hade a DC motor and the results were the same in different tests we did. We will include more accurate motors for future versions of this sensors and we will compare it.

Comment: 12 shows the workflow of the proposed system.  Is the diagram associated with the Python application? Are the blocks related with the function implemented in Python?

Answer : Yes, the phyton application is in charge of checking rules taking into account the received data. Additionally. It calls other functions to, for example, draw the map.

We have added this information to be sure that no doub appears regarding to the diagram interpretation. The following text has been added before figure 12: “Figure 12 shows the workflow diagram of our system. The diagram shows the workflow of our phyton application which is in charge of checking rules taking into account the received data. Additionally. It calls other functions to, for example, draw the map.”

Comment: What is eqfeed_callback()? In the conditions “i>=length …” i should be lower case. In some blocks is written coordenate instead of coordinates.

Answer : eqfeed_callback () creates a file where the coordinates of the nodes that exceed the values imposed by the 30-30-30 rule will be saved. It gets the values of the JSON objects from the "data.json" file. It defines a marker that indicates fire if the value of the CO2 concentration exceeds the 700 ppm threshold and displays an alert by screen. This function also defines the information windows that will be displayed by clicking on the markers and their format. Finally, the function is in charge of painting both the markers and the heat map, in case the defined thresholds have been exceeded.

Figure 12 has been modified according to the reviewer´s comments.

The following text has been added before figure 12:

“eqfeed_callback(): it creates a file where the coordinates of the nodes that exceed the values imposed by the 30-30-30 rule will be saved. It gets the values of the JSON objects from the "data.json" file. It defines a marker that indicates fire if the value of the CO2 concentration exceeds the 700 ppm threshold and displays an alert by screen. This function also defines the information windows that will be displayed by clicking on the markers and their format. Finally, the function is in charge of painting both the markers and the heat map, in case the defined thresholds have been exceeded. “

Comment: According to what stated in Line 495 and in the conclusion section, experimental tests have been carried out with LoRa signals characterized by SF equal to 10; on the contrary, reported results and consumption considerations are associated with SF=7 (e.g. Fig.17 and line 602).

Answer : The SF used is 7. It was an error in the text.

Comment: Description of results about temperature tests does not correspond to what shown in Fig.20 Why is there a discontinuity in the relative humidity curve? How do the authors emulate fire in tests with CO2 sensor? Also at night?

Answer : We have corrected the comments related to the temperatura.

The discuntinuity in Relative humity is due to the decreasing of wind speed. The tests wer performed near to the coast and the presence of dew is common. When the day is windy the dew decreases. When wind speed decreases the dew tends to increase. However 30% of RH is a very dry environment.

We used a kitchen smoke pipe that burns small pieces of wood and generates smoke. The tests with sensors were carried out in a natural park so no fire or similar were permited. The following text has been added before figure 23: “ To generate de smoke, we have used a kitchen smoke pipe.”

Comment: Line 568 FR should be RF?

Answer : Yes. It has been corrected.

Comment: Table 4 Consumptions of device should be in mA, not mAh

Answer :  Yes. It has been corrected.

Comment: According to what states Tab. 5, the most consuming operation is the measurement of sensor and not Tx LoRa as reported in the following lines 594-595

Answer : Yes, it is true. The comment has been revised as follow:

“As it can be seen, the mode that consumes more current is the state where the sensors are collecting the data from the environment which is almost four times higher than the process of wirelessly transmitting the message with the information.”

Comment: Line 600 It is better to write 1680000 ms

Answer : Yes. It has been corrected.

Comment: Calculations for the definitions of consumptions of Tx LoRa have to be updated with the correct SF; if this is the case, consumption should be 8 times higher.

Answer : The SF used is 7. It was an error in the text.

Comment: In Eq. 8 e 9, the same consumption of 198700mAms is given; I think this is a “copy-and-past” error

Answer : Yes, it was a “copy-and-past” error. Equations have been corrected as follow

Comment: The authors do not taken into account self-discharge in determining the lifetime of the battery

Answer : We do not take into account the self-discharge, but we know that the battery’s lifetime will be slightly lower.

We have added this note after table 6, just to be clear how these values have been calculated:

“(without taking into account the self-discharge. In this sense, the battery’s lifetime will be slightly lower).”

Round 2

Reviewer 3 Report

The authors have responded positively to all the raised comments.

I believe there is still some problem with the visibility of the green leaf in Figs. 13 and 15: when printed, it is barely separable from the background and, thus, almost invisible.

I suggest to double-check again the English language and style in the text added during the revision phase.

Author Response

Answer : Thanks for reviewing our paper. Thank you for your pertinent comments and detailed observations. These were really useful to improve the quality of our paper. Our responses are in line, starting with “Answer:”.

Comment: I believe there is still some problem with the visibility of the green leaf in Figs. 13 and 15: when printed, it is barely separable from the background and, thus, almost invisible.

Answer: Figures 13 and 15 has been replaced by the following ones where the fourth icon is now visible.

Figure 13. Website showing 4 nodes.

Figure 15. Website showing 4 nodes and the measured data.

Comment: I suggest to double-check again the English language and style in the text added during the revision phase.

Answer: As reviewer suggested, we have performed a double-check of the English language and style. The corrections have been marked in red on the paper.

This manuscript is a resubmission of an earlier submission. The following is a list of the peer review reports and author responses from that submission.

Round 1

Reviewer 1 Report

In the abstract the sentence "To evaluate the fire risk the system implement the 30-30-30 rule. Data from nodes are stored and processed in a The Thing Network (TTN) server that send data in .json format to a web site ..." need to be improved: what is the 30-30-30 rule? .json format is quite irrelevant for the purpose of the implementation and shoudl not be in an abstract.

The sentence "LoRaWAN specifications were developed by the French company Cycleo" is apparently false. The authors confuse LoRa and LoRaWAN in the paper.

Regarding the description of LoRa and LoRaWAN the authors should just point to some references. The description does not add any value to the paper.

3.2.2. Standardization paragraph title is misleading; the authors confuse standardization and regulation.

Table 1 is misleading. WHat is the payload size? The time on air heavily depends on that.

3.3. Comparison with other technologies is irrelevant.

RSSI measurements are irrelevant (just going on Google Scholar and one would see how many papers have dealt with that topic much more in depth).

Overall the conclusion that with a coverage range of 4km per gateway you can monitor the fire in the forests is wrong: how many gateways would one need for a coverage of a real forest? and how to connect the gateways to the internet?

Reviewer 2 Report

The paper proposes the use of LoRa technology to wirelessly collect a variety of sensor information at a centralized server and assess the risk level or the presence of a forest fire by analyzing the received data. The proposed system is tested in a real environment and the possible area coverage by using a single gateway is estimated.

The proposed method, including the background about the LoRa technology, the device/sensor and microcontroller selection, the network setup instructions, and finally the discussions on the obtained results are valuable and well drafted. Overall, it appears that the paper mainly suggests a deployment and utilization of the readily available sensors and technology in a specific use case. This is closer to a project report than a research publication. However, I leave this to the discretion of the editor to decide whether this type of contribution is acceptable for publication in this journal. Regardless, I have some concerns/comments as listed below that I think will help improving the quality of the paper by fixing them before being ready for publication:

The title suggests “Low-Cost Network for …”, however, I don’t see much of contribution in designing, deploying or proposing a new networking paradigm in the paper. It rather looks as a data collection of randomly distributed off-the-shelf available nodes/end-devices. At minimum, I suggest to either correct the title or add some contribution to the networking side of the problem. In the introduction the authors have done a great job in motivating the problem, but I wonder why the statistics are restricted to specific country, i.e., Spain. Needless to say, the forest fires are not limited to a specific geographic location, therefore, I’d rather emphasize on the broader impact of the research by adding some global statistics. In the introduction, the authors mention Sigfox as a possibility, but then refer to LoRa to have more popularity. Isn’t it the case that Sigfox has a much better coverage specifically in Europe with a well-deployed network? Also, how does the authors compare these two technologies? As they are both low power and long range, what are the advantages of one over the other, and why is the reason that LoRa is preferred for this application by the authors. Maybe mentioning things like: robustness to noise and environment changes (multipath fading, etc.) of the CSS PHY makes LoRa an ideal case for deployment in forests, or other differences between these two low power technologies have made the authors to motivate the utilization of LoRa. In any case, I suggest that the authors shed some light on their technology choice which will help the readers to better understand the benefits of one over another and use this information for their future research. Section III has very good information, but if I were to make the paper shorter, I would cut most of this section and would probably only keep the comparison table II. All the rest of the information can be found online, and you can refer the authors to the corrects sources, e.g., the LoRa Alliance website and documentations. Also, in Section IV, all the detailed information about different sensors that are used are appreciated for someone who wants to replicate the system, but I would rather only include the part numbers. The interested authors can easily find the rest of the information in online datasheets. Personally, I think theses information (operating voltage of the sensor, or the calibration method of the CO2 sensor, etc.) are irrelevant, specifically as long as they are not used in any of the analysis presented in the paper, neither do they have any value in presenting the results or conclusions. Similarly, in Section V, a large portion (essentially all the information before Figure 15) is just about the network settings to a specific web server/host. It all looks like a very well written tutorial or instruction manual, but these are easily accessible to the readers through websites like TTN or the similar. It might be the case that this journal does not limit the authors with the available space, i.e., number of pages, nonetheless, it’s good practice that the authors only include the crucial information about the system model, methodology, analysis, results, conclusion, etc., and dedicate the main part (majority of the paper) to their unique proposal, contributions and the outcomes. What does Free Prop. Model in Figure 21 refer to? I assume it is Free Propagation Model, but I suggest to explain this with more details. In the explanation of Figure 21 (or 22) the authors suggest: “Regarding to the SNR values, they are so dispersed. This behavior is due to the influence of buildings and obstacles.” The dispersity of the SNR is not necessarily caused by the buildings, please be clearer and more concise in your reasoning. The conclusion section is rather short and lacks enough elaboration around the findings, outcomes, and explanation of the figures. While the authors can cut large parts from sections III, IV and V, I believe the last two sections, i.e., VI and VII still need to be improved and enhanced as this is the actual contribution of the authors and contains the unique methodology and the consequential results. Temperature of which one of the sensors is shown in Figure 22? Isn’t it the case that you have deployed multiple sensors? Is this the average or the collected data or that of a specific sensor? In any case, what is the value in including this figure and not showing the contribution of the collected data in the results? In other words, only knowing the temperature of a node doesn’t have any value by itself, and outside of an appropriate context. Similarly, in Fig. 23, the humidity, and Fig. 24, the wind speed, so on so forth… I believe the authors have processed this data and based on the analysis they are able to detect the probability of a fire, e.g., how do you compute the values shown in Fig. 16 and Fig. 18, or how is the notification in Fig. 17 triggered. In general, I see a lack of connection between the actual meaningful results (presented in the previous section) and the individual readings of different sensors (that shows up later on in the paper). Perhaps, rearranging some of the things in these two sections and connecting them together will make it more useful and easier for the readers to follow. How much energy does this network consume? And what is the cost of deployment? How can it be justified that this is a “Low Power” and “Low Cost” system? I think these are the key questions that needs to be answered in the paper to make it impactful and beneficial for the prospective readers. I don’t think, the paper is about building a coverage map, it rather claims to be about forest fire detection and prevention. However, the results more elaborate on the former that the later. I suggest, to either, change the motivation and title to the former, or reflect/elaborate more of the later in the manuscript. The suggested future work seems to be a well-studied as the authors have already provided references. My concern/question would be, what will be the novelty/contribution in the proposed future research? In conclusions, the authors suggest that: “From our results, we can extract that it is possible to develop low-cost LoRa-based nodes to implement an autonomous system for evaluating the fire risk of an area and for detecting the presence of a fire when it occurs in real-time.” My repeated concern is, the manuscript does not seem to consider any cost analysis to justify the claim of proposing a low-cost system. At the end, I’d like to mention that your work is really valuable, and the experimental setup and the amount of effort that you put through to obtain and present the results are significant. Therefore, please consider all the above comments as hopefully productive suggestions with sole intention of helping to improve the impact of your work. I hope you find them useful and that it helps to improve the quality of your work.

Round 2

Reviewer 1 Report

The English should be improved in the reviewed version: for example "bigger Data Transfer Rate." is not English.

The authors failed completely to answer the major question "Overall the conclusion that with a coverage range of 4 km per gateway you can monitor the fire in the forests is wrong: how many gateways would one need for a coverage of a real forest? and how to connect the gateways to the internet?"

A forest - in the reviewer's opinion - is much more than just 4 km. Ho to connect gateways in the middle of a real forest to the internet is still not addressed 

Reviewer 2 Report

The authors have tried to address some of the comments, but it seems to be done in a rush. Grammatical errors are increased rather than being improved. The power consumption calculation that is added in the revised version does not make any sense. 10,000 mAh battery for a single node?! This is twice the capacity of a laptop battery! You can run an 8-core processor with a large screen for 15 hrs or so... How come a LoRa node (using supposedly a very low power technology) need that large of a battery and lasts only 40 hours?! Typical sensors that you use, and considering the transmission power consumption of LoRa, you should be able to have thousands of transmissions with few hundred mAh battery. Please reconsider this section, and if the numbers are correct, then reconsider your design...